# Rap2 and TNIK control Plexin-dependent tiled synaptic innervation in *C. elegans*

Xi Chen[1], Akihiro CE Shibata[2†], Ardalan Hendi[1†], Mizuki Kurashina[1], Ethan Fortes[1], Nicholas L Weilinger[3], Brian A MacVicar[3], Hideji Murakoshi[2], Kota Mizumoto[1]*

[1]Department of Zoology, The University of British Columbia, Vancouver, Canada; [2]Supportive Center for Brain Research, National Institute for Physiological Sciences, Okazaki, Japan; [3]Department of Psychiatry, The University of British Columbia, Vancouver, Canada

**Abstract** During development, neurons form synapses with their fate-determined targets. While we begin to elucidate the mechanisms by which extracellular ligand-receptor interactions enhance synapse specificity by inhibiting synaptogenesis, our knowledge about their intracellular mechanisms remains limited. Here we show that Rap2 GTPase (*rap-2*) and its effector, TNIK (*mig-15*), act genetically downstream of Plexin (*plx-1*) to restrict presynaptic assembly and to form tiled synaptic innervation in *C. elegans*. Both constitutively GTP- and GDP-forms of *rap-2* mutants exhibit synaptic tiling defects as *plx-1* mutants, suggesting that cycling of the RAP-2 nucleotide state is critical for synapse inhibition. Consistently, PLX-1 suppresses local RAP-2 activity. Excessive ectopic synapse formation in *mig-15* mutants causes a severe synaptic tiling defect. Conversely, overexpression of *mig-15* strongly inhibited synapse formation, suggesting that *mig-15* is a negative regulator of synapse formation. These results reveal that subcellular regulation of small GTPase activity by Plexin shapes proper synapse patterning in vivo.
DOI: https://doi.org/10.7554/eLife.38801.001

*For correspondence:
mizumoto@zoology.ubc.ca

†These authors contributed equally to this work

## Introduction

During nervous system development, various instructive and repulsive signaling cues cooperatively direct neurons to form chemical synapses with their appropriate targets. Studies have identified some molecules and elucidated their downstream mechanisms that instruct synaptogenesis such as FGF, Ephrin/Eph, Ig-family of cell adhesion molecules (IgCAMs) and synaptic cell adhesion molecules (SynCAMs) (*Shen and Bargmann, 2003*; *Shen et al., 2004*; *Dabrowski and Umemori, 2016*; *Dabrowski et al., 2015*; *Feng et al., 2000*; *Kayser et al., 2008*; *Terauchi et al., 2010*; *Yamagata et al., 2003*). Several axon guidance cues and their receptors also play critical roles to inhibit synapse formation (*Inaki et al., 2007*; *Klassen and Shen, 2007*; *Poon et al., 2008*). Semaphorins (Sema) and their receptors, Plexins, are two conserved families of molecules that have a well-established function to repel axons during development (*Kolodkin et al., 1993, 1992*; *Negishi et al., 2005a*; *Tran et al., 2007*) and play prominent roles contributing to immune system, cardiovascular development and cancer regulation (*Epstein et al., 2015*; *Neufeld et al., 2005*; *Takamatsu and Kumanogoh, 2012*).

In addition to its function as a long-range axon guidance cue during neuronal development, Sema/Plexin signaling plays a critical role as a negative regulator of synapse formation. The role of Sema/Plexin signaling to inhibit synapse formation was first observed in Drosophila, where ectopic expression of Sema2a causes elimination of specific neuromuscular junctions (*Matthes et al., 1995*). In mammals, Sema3E/PlexinD1 specifies sensory-motor connections (*Pecho-Vrieseling et al., 2009*). Secreted Sema3F locally inhibits spine development through its receptors PlexinA3 and Neuropilin-2

**eLife digest** Genes do more than just direct the color of our hair or eyes. They produce proteins that are involved in almost every process in the body. In humans, the majority of active genes can be found in the brain, where they help it to develop and work properly – effectively controlling how we move and behave.

The brain's functional units, the nerve cells or neurons, communicate with each other by releasing messenger molecules in the gap between them, the synapse. These molecules are then picked up from specific receptor proteins of the receiving neuron.

In the nervous system, neurons only form synapses with the cells they need to connect with, even though they are surrounded by many more cells. This implies that they use specific mechanisms to stop neurons from forming synapses with incorrect target cells. This is important, because if too many synapses were present or if synapses formed with incorrect target cells, it would compromise the information flow in the nervous system. This would ultimately lead to various neurological conditions, including Autism Spectrum Disorder.

In 2013, researchers found that in the roundworm *Caenorhabditis elegans*, a receptor protein called Plexin, is located at the surface of the neurons and can inhibit the formation of nearby synapses. Now, Chen et al. – including one author involved in the previous research – wanted to find out what genes Plexin manipulates when it stops synapses from growing. Knowing what each of those genes does can help us understand how neurons can inhibit synapses.

The results revealed that Plexin appears to regulate two genes, Rap2 and TNIK. Plexin reduced the activity of Rap2 in the neuron that released the messenger, which hindered the formation of synapses. The gene TNIK and its protein on the other hand, have the ability to modify other proteins and could so inhibit the growth of synapses. When TNIK was experimentally removed, the number of synapses increased, but when its activity was increased, the number of synapses was strongly reduced.

These findings could help scientists understand how mutations in Rap2 or TNIK can lead to various neurological conditions. A next step will be to test if these genes also affect the formation of synapses in other species such as mice, which have a more complex nervous system that is structurally and functionally more similar to that of humans.

DOI: https://doi.org/10.7554/eLife.38801.002

in hippocampal granule cells (*Tran et al., 2009*). Sema5A/PlexinA2 signaling inhibits excitatory synapse formation in dentate granule cells (*Duan et al., 2014*). Sema5B diminishes synaptic connections in cultured hippocampal neurons (*O'Connor et al., 2009*). However, little is known about the intracellular mechanisms through which Sema/Plexin signaling inhibits synapse formation.

The cytoplasmic domain of Plexin contains a GAP (GTPase-activating protein) domain that inactivates small GTPases (*Oinuma et al., 2004*; *Rohm et al., 2000*). Upon activation by Semaphorins, Plexins repel axon outgrowth by inhibiting R-Ras (*Negishi et al., 2005b*; *Hota and Buck, 2012*; *Tasaka et al., 2012*). Recent biochemical and structural analyses demonstrated that the GAP domain of mammalian PlexinA3 is specific for Rap GTPases, which belong to the Ras family of GTPases and regulate the actin cytoskeleton (*Wang et al., 2012*, *2013*). PlexinA3 dimerization by Semaphorin activates its GAP domain, thereby inhibiting Rap1 from inducing neurite retraction. Drosophila PlexA and zebrafish PlexinA1 promote remodeling of epithelial cells by inhibiting Rap1 GTPase during wound healing (*Yoo et al., 2016*). Another Rap GTPase, Rap2, can inhibit neurite outgrowth (*Kawabe et al., 2010*). Similar to Sema/Plexin signaling, Rap GTPases regulate synapse formation and function. Rap2 negatively regulate spine number in cultured hippocampal neurons (*Fu et al., 2007*). Rap1 and Rap2 regulate synaptic activity by removing AMPA receptors from spines during long-term depression and depotentiation, respectively (*Zhu et al., 2002*, *Zhu et al., 2005*). While the GAP domain of Plexin is critical to inhibit synapse formation (*Duan et al., 2014*; *Mizumoto and Shen, 2013a*), we still do not know whether Plexin regulates synapse patterning via Rap GTPases at presynaptic sites.

In *Caenorhabditis elegans*, Sema/Plexin signaling functions in vulva formation and male ray development (*Dalpé et al., 2005*, *2004*; *Fujii et al., 2002*; *Ikegami et al., 2004*; *Liu et al., 2005*;

*Nakao et al., 2007*; *Nukazuka et al., 2008*, *2011*). Using this model system, we previously reported that Sema/Plexin signaling in the nervous system mediates a critical inter-axonal interaction for the tiled synaptic innervation of two DA-class cholinergic motor neurons (DA8 and DA9) (*Mizumoto and Shen, 2013a*). Cell bodies of nine DA neurons in *C. elegans* reside in the ventral nerve cord, sending dendrites ventrally and axons dorsally to form *en passant* synapses onto the dorsal body wall muscles. Even though axons of DA neurons show significant overlap, each motor neuron forms synapses onto muscles within specific sub-axonal domains, which do not overlap with those from neighboring DA neurons. This unique synaptic innervation creates tiled synaptic patterns along the nerve cord (*White et al., 1986*).

Tiled synaptic innervation occurs within most motor neuron classes and may contribute to the sinusoidal locomotion pattern of *C. elegans* (*White et al., 1986*). Using a combination of two fluorescent proteins (GFP and mCherry) fused with the presynaptic vesicle protein, RAB-3, and two tissue specific promoters (*Figure 1*) (*Mizumoto and Shen, 2013a*), we can visualize this synaptic tiling between DA8 and DA9 neurons. We reported that PLX-1 localizes at the anterior edge of the DA9 synaptic domain in axon-axon interactions in a Semaphorin-dependent manner, where it locally inhibits formation of the presynaptic specialization via its GAP domain. Loss of *Semas* or *plx-1* causes anterior expansion of DA9 synaptic domain and posterior expansion of DA8 synaptic domain. This result indicates loss of inter-axonal interactions between DA8 and DA9 neurons. Consistently, Tran et al., also observed excess dendritic spine formation, specifically within the region close to the cell body, in the *plexin* knockout mouse (*Tran et al., 2009*). These findings suggest a conserved mechanism by which Sema/Plexin locally inhibits synapse formation.

We previously reported that *let-60/KRas* gain-of-function mutants showed very mild synaptic tiling defects. Since mammalian Plexin acts as a RapGAP, we hypothesized that Rap GTPase is the major downstream effector of PLX-1 to regulate synaptic tiling of DA neurons. Here, we report that *rap-2*, a *C. elegans* ortholog of human Rap2A, and its effector kinase *mig-15* (TNIK: Traf2- and Nck-interacting kinase) act genetically downstream of *plx-1* to regulate synaptic tiling. PLX-1 delineates the border of synaptic tiling by locally inhibiting RAP-2 along the DA9 axon. We also discovered an unexpected role for *mig-15* in inhibiting synapse formation. Our results reveal the mechanism underlying Plexin signaling to form fine synaptic map connectivity.

## Results

### *rap-2* functions downstream of *plx-1* to regulate synaptic tiling

Three Rap genes exist in the *C. elegans* genome (*rap-1*, *rap-2* and *rap-3*). To delineate which Rap GTPase functions downstream of PLX-1 in synaptic tiling between DA8 and DA9 neurons, we first examined the expression patterns of all three *rap* genes (*Figure 1—figure supplement 1*). Among them, only *rap-2*, an ortholog of mammalian Rap2a, was expressed in motor neurons including DA8 and DA9, while *rap-1* and *rap-3* were not expressed in these cells (*Figure 1—figure supplement 1*). In wild type animals, synaptic domains of DA8 and DA9 neurons did not show significant overlap, creating tiled synaptic innervation (*Figure 1A and G*). In the *plx-1(nc36)* null mutant, synaptic domains of DA8 and DA9 expanded posteriorly and anteriorly, respectively. As a result, synaptic domains of these neurons overlapped significantly (*Figure 1B and G*).

Since the intracellular domain of Plexin contains a RapGAP domain, we hypothesized that RAP-2 preferentially exists in a GTP-bound form in the *plx-1* mutants. The G12V mutant is widely used as a constitutively GTP-form of small GTPases including mammalian Rap2A and *C. elegans* RAP-1 (*Kawabe et al., 2010*; *Pellis-van Berkel et al., 2005*). Expression of a constitutively GTP-bound form of *rap-2(G12V)* under the A-type neuron specific promoter, P*unc-4*, elicited a similar synaptic tiling defect as *plx-1* mutants (*Figure 1C and G*). Expression of wild type *rap-2* under the *unc-4* promoter did not affect the synaptic tiling pattern, suggesting that G12V mutation but not over-expression of *rap-2* caused the synaptic tiling defect (*Figure 1G*). We then generated *rap-2(G12V)* mutants using CRISPR/Cas9 genome editing. We observed the same synaptic tiling defects in three independent *rap-2(G12V)* mutant alleles (*miz16*, *miz17* and *miz18*) as in *plx-1* mutants (*Figure 1D and G* and *Figure 1—figure supplement 2*). We found a comparable level of gene expression among all three *rap-2(G12V)* mutants to wild type *rap-2* using RT-qPCR (*Figure 1—figure supplement 2*). These

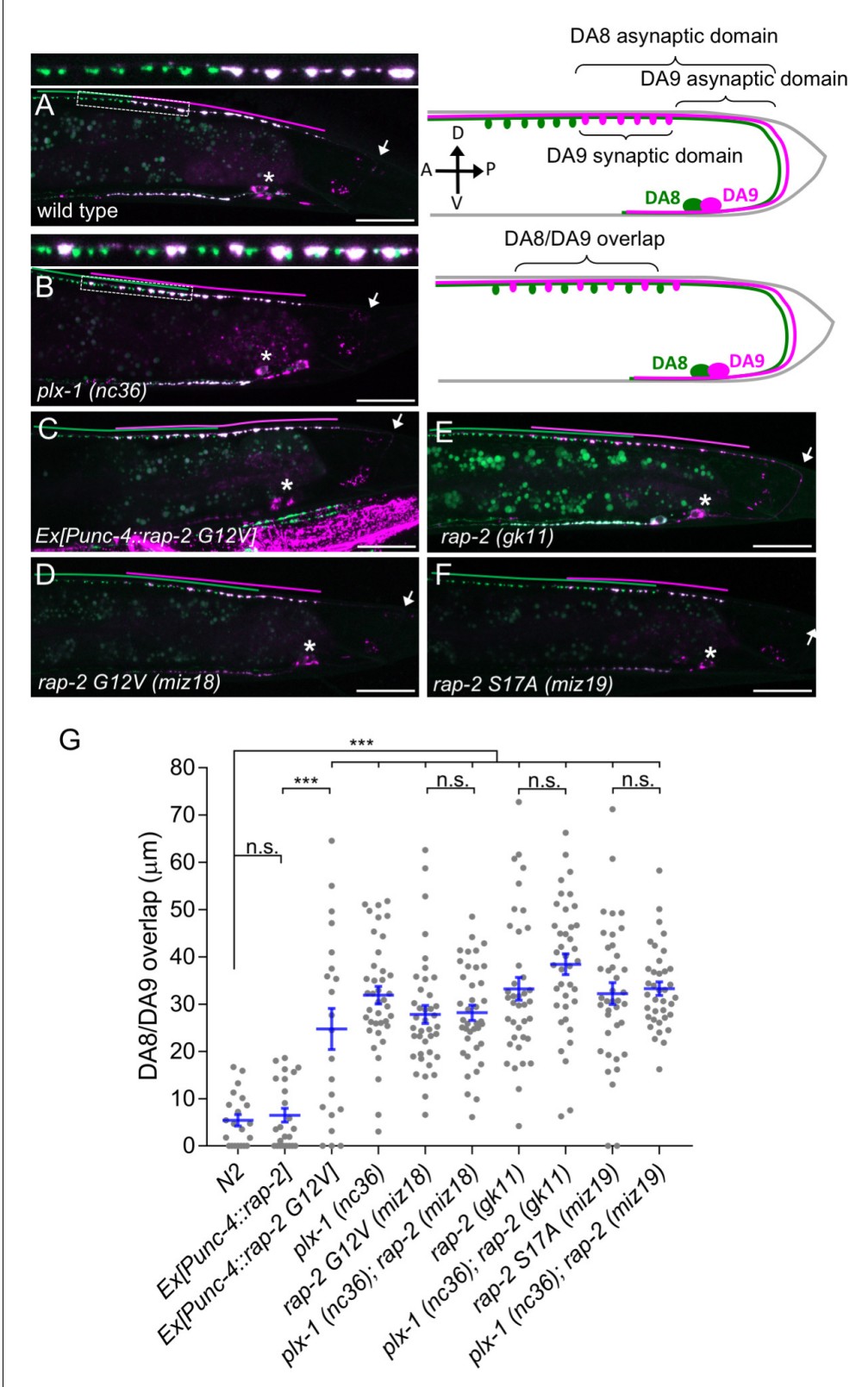

**Figure 1.** Gain- and loss-of-function *rap-2* mutants show synaptic tiling defects. (**A** and **B**) Representative image of synaptic tiling of wild type (**A**) and *plx-1* mutant (**B**) animals. Images show *wyIs446* marker to label synapses in DA8 (GFP::RAB-3) and DA9 (GFP::RAB-3+mCherry::RAB-3). Dotted box represents the magnified images from A and B of the synaptic tiling border. Schematics of DA8 (green) and DA9 (magenta) neurons with parameters for analysis
*Figure 1 continued on next page*

*Figure 1 continued*

shown on the right. (C–F) Representative images of *wyIs446* strains with the following genotypes: *rap-2(G12V)* overexpression in DA neurons (C), *rap-2 G12V (miz18)* (D), *rap-2 null (gk11)* (E) and *rap-2 S17A (miz19)* (F). Synaptic domains from DA8 and DA9 are highlighted with green and magenta lines, respectively. Asterisks: DA9 cell body. Arrows: dorsal commissure of DA9. Scale bars: 20 μm. (G) Quantification of overlap between DA8 and DA9 synaptic domains. Each dot represents a single animal. Blue bars indicate mean ± SEM. n.s.: not significant; ***p<0.001.

DOI: https://doi.org/10.7554/eLife.38801.003

The following figure supplements are available for figure 1:

**Figure supplement 1.** *rap-1* and *rap-3* are not involved in synaptic tiling of DA8 and DA9.
DOI: https://doi.org/10.7554/eLife.38801.004
**Figure supplement 2.** Synaptic tiling requires both GTP- and GDP-bound forms of RAP-2.
DOI: https://doi.org/10.7554/eLife.38801.005
**Figure supplement 3.** Co-localization between RAB-3 and the active zone markers, CLA-1 and UNC-10.
DOI: https://doi.org/10.7554/eLife.38801.006

results confirm that the *rap-2(G12V)* mutation itself, not changes in gene expression, underlie the synaptic tiling defect in *rap-2(G12V)* mutants.

Surprisingly, the null mutant of *rap-2(gk11)* also showed the same synaptic tiling defect, as did the constitutively GTP-form of *rap-2(G12V)* mutants (**Figure 1E and G**). This result suggests that synaptic tiling requires both the GTP- and GDP-bound forms of RAP-2. Indeed, the constitutively GDP-bound form of *rap-2* mutants (S17A: *miz19, miz20*) showed the identical synaptic tiling defect as *rap-2(G12V)* and *rap-2(gk11)* mutants (**Figure 1F and G** and **Figure 1—figure supplement 2**). These results suggest that the cycling between GTP- and GDP-forms of RAP-2 is critical to regulate the spatial patterning of synapses.

Similar to *plx-1* mutants, the synaptic tiling defect in all *rap-2* mutants is caused by both the posterior expansion of DA8 synaptic domain and the anterior expansion of the DA9 synaptic domain (**Figure 1—figure supplement 2**). However, none of *rap-2(G12V)*, *rap-2(S17A)* and *rap-2(gk11)* mutants enhanced or suppressed the synaptic tiling defect in *plx-1* mutants (**Figure 1G**). These results suggest that *plx-1* and *rap-2* function in the same genetic pathway. Consistent with the expression patterns of *rap-1* and *rap-3*, neither *rap-1(pk2082)* nor *rap-3(gk3975)* null mutants showed significant synaptic tiling defects by themselves and did not enhance the synaptic tiling defect in *rap-2(gk11)* null mutants (**Figure 1—figure supplement 1**). All mCherry::RAB-3 puncta co-localized with active zone markers, CLA-1 (**Xuan et al., 2017**) and UNC-10/Rim (**Wu et al., 2013**) in the DA9 neurons of *plx-1(nc36)* and *rap-2(gk11)* mutants, suggesting that RAB-3 puncta represent *bona fide* synapses (**Figure 1—figure supplement 3**). Taken together, these data indicate that *plx-1* and *rap-2* act in the same genetic pathway for synaptic tiling in DA neurons.

## RAP-2 functions cell autonomously in DA neurons

We next determined the cellular location for *rap-2* function. Since *rap-2(gk11)* null mutants showed a synaptic tiling defect, we conducted tissue specific rescue experiments using tissue-specific promoters as previously described (**Mizumoto and Shen, 2013a**). Expression of *rap-2* in the post-synaptic body wall muscle cells under the *hlh-1* promoter or in another class of cholinergic motor neurons in the dorsal nerve cord (DB neurons) under the truncated *unc-129* promoter did not rescue the synaptic tiling defect in *rap-2(gk11)* animals (**Figure 2A, B and G**). However, DA neuron-specific expression using the *unc-4c* promoter strongly rescued the synaptic tiling defect (**Figure 2C and G**). DA9-specific expression of *rap-2* under the *mig-13* promoter partially rescued the synaptic tiling defect (**Figure 2E and G**). DA9-specific expression of *rap-2* rescued the phenotype of anterior expansion of the DA9 synaptic domain but not the posterior expansion of DA8 synaptic domain (**Figure 2H and I**). These results suggest that *rap-2* regulates synapse patterning in a cell-autonomous manner. In contrast to the DA9-specific rescue experiment in *rap-2* mutants, DA9-specific expression of *plx-1* cDNA was sufficient to rescue synaptic defects in both DA9 and DA8 (**Mizumoto and Shen, 2013a**) (see Discussion).

We also observed that expression of human Rap2a in DA neurons rescued the synaptic tiling defect of *rap-2* mutants, suggesting the function of *rap-2* in synapse patterning is conserved across

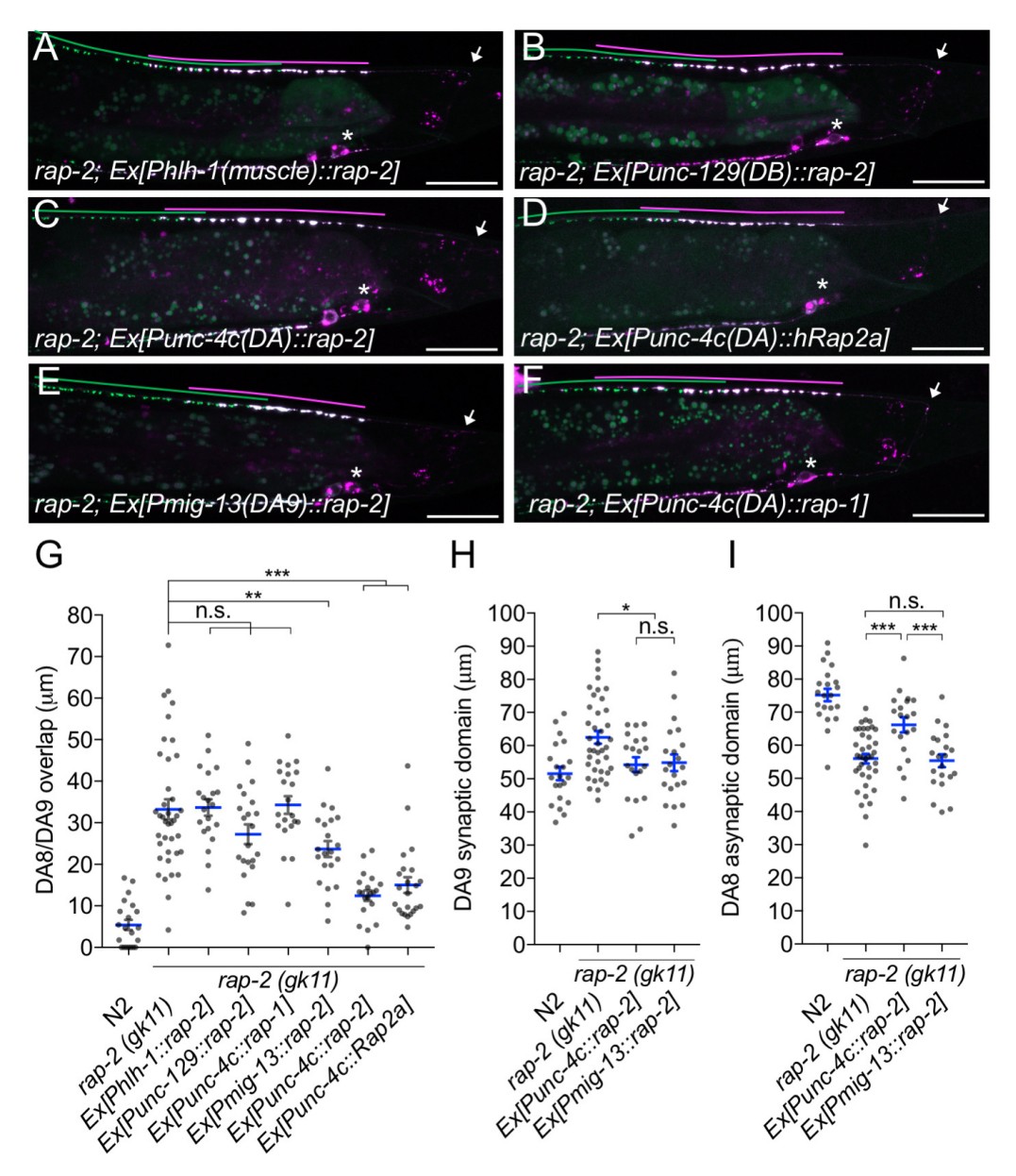

**Figure 2.** *rap-2* functions in DA neurons. (A–F) Representative images of *rap-2; wyIs446* animals expressing *Phlh-1::rap-2* (A), *Punc-129::rap-2* (B), *Punc-4c::rap-2* (C), *Punc-4c::Rap2a (human)* (D), *Pmig-13::rap-2* (E) and *Punc-4c;;rap-1* (F). Synaptic domains of DA8 and DA9 are highlighted with green and magenta lines, respectively. Asterisks: DA9 cell body. Arrows: dorsal commissure of DA9. Scale bars: 20 μm. (G–I) Quantification of DA8/DA9 overlap (G), DA9 synaptic domain (H) and DA8 asynaptic domain (I). Each dot represents a single animal. Blue bars indicate mean ± SEM. n.s.: not significant; ***p<0.001; **p<0.01; *p<0.05.

DOI: https://doi.org/10.7554/eLife.38801.007

species (*Figure 2D and G*). Previous work suggested a partial functional redundancy between *rap-1* and *rap-2* in *C. elegans* (*Pellis-van Berkel et al., 2005*). However, we found that *rap-1* expression in DA neurons did not rescue the synaptic tiling defect of *rap-2* mutants, suggesting functional diversity between *rap-1* and *rap-2* (*Figure 2F and G*). Taken together, we conclude that *rap-2* functions cell autonomously in DA neurons to regulate synaptic tiling.

## RAP-2 activity is spatially regulated by PLX-1

Previously, we demonstrated that PLX-1::GFP is localized at the anterior edge of the DA9 synaptic domain, where it negatively regulates synapse formation through its cytoplasmic GAP domain (*Figure 3A and E*) (*Mizumoto and Shen, 2013a*). In the *rap-2(gk11)* mutant background, we observed no change in PLX-1::GFP localization but did observe ectopic synapses in the axonal region anterior to the PLX-1::GFP domain (*Figure 3B and F*). This result is consistent with our hypothesis that *rap-2* acts downstream of *plx-1* to regulate synaptic tiling. Together with our finding that synaptic tiling requires both GTP- and GDP-bound forms of RAP-2, we speculate that PLX-1 acting at the anterior edge of the DA9 synaptic domain regulates the spatial activity of RAP-2 along the axon.

We then sought to determine the spatial distribution of GTP-RAP-2 in DA9 axon. We conducted Fluorescence Lifetime Imaging Microscopy (FLIM)-based FRET (Förster Resonance Energy Transfer) measurements using EGFP-Rap2A (human) and mRFP-RalGDS(RBD: Ras Binding Domain)-mRFP (*Yasuda et al., 2006*). As RalGDS-RBD specifically binds to GTP-Rap2 but not GDP-Rap2 (*Ohba et al., 2000*), FRET from EGFP-Rap2A to mRFP-RalGDS(RBD)-mRFP can be used as a readout of Rap2 activity. We detected FRET signal as a change of GFP fluorescence lifetime (*Figure 4A*). In HeLa cells, we observed a shorter lifetime of constitutively bound GTP construct EGFP-Rap2A(G12V) compared to GDP-bound EGFP-Rap2A(S17A), indicating that the FRET sensor can detect the nucleotide state of Rap2A (*Figure 4B and C*). Due to the low expression of *C. elegans* RAP-2 constructs in HeLa cells, we were not able to test whether the mammalian FRET sensor can detect *C. elegans* RAP-2 activity (data not shown).

We then expressed EGFP-Rap2A and mRFP-RalGDS(RBD)-mRFP FRET sensors in DA9 neurons in *C. elegans*. As human Rap2a rescued the synaptic tiling defect of *rap-2(gk11)* mutants (*Figure 2G*), we reasoned that the activity pattern of human Rap2A should recapitulate that of endogenous RAP-2. We indeed observed lower Rap2A activity at the anterior edge of the DA9 synaptic domain compared to within the synaptic domain (*Figure 4D and F*). This observation is consistent with the localization of PLX-1::GFP at the anterior edge of DA9 synaptic domain (*Figure 3A*) (*Mizumoto and Shen, 2013a*). Local inhibition of Rap2a activity was strongly diminished in the *plx-1* mutant background (*Figure 4D and F*). Higher Rap2 activity in the synaptic region could simply indicate the presence of synapses within the synaptic domain, rather than Rap2 inactivation by Plexin at the anterior edge of the synaptic domain. To exclude this possibility, we examined Rap2 activity in *unc-104/Kif1A* mutants, which show no synapses are formed in DA9 axon (*Ou et al., 2010*). We showed previously that PLX-1::GFP localization to the synaptic tiling border was independent of synapses, since it was unaffected in *unc-104/Kif1A* mutants (*Mizumoto and Shen, 2013a*). In *unc-104* mutants, we observed the same local inhibition of Rap2A activity at the putative synaptic tiling border, but not in *unc-104; plx-1* double mutants (*Figure 4E and G*), indicating that Plexin controls local Rap2 activity independent of synapses.

To understand that this local Rap2 inactivation at the synaptic tiling border depends on the localized RapGAP activity of PLX-1, we examined the rescue activity of two PLX-1 mutant constructs, PLX-1(RA) and PLX-1(ΔSema), neither of which rescued the synaptic tiling defect of *plx-1* mutants (*Mizumoto and Shen, 2013a*). PLX-1(RA) is a GAP-deficient mutant but localizes normally at the anterior edge of the DA9 synaptic domain. PLX-1(ΔSema) contains intact GAP domain but cannot be activated by the endogenous ligand and shows diffused localization due to deletion of the extracellular SEMA domain (*Mizumoto and Shen, 2013a*). We observed no local Rap2 inactivation in *plx-1* mutant animals expressing these mutant PLX-1 constructs in DA9, while expression of wild type PLX-1 cDNA rescued local Rap2 inactivation at the anterior edge of the DA9 synaptic domain (*Figure 4H*).

While we do not fully exclude the possibility that PLX-1 indirectly regulates local Rap2 activity, these data taken together with the biochemical evidence that mammalian Plexin acts as RapGAP (*Wang et al., 2013*, *2012*) strongly suggests that Plexin localized at the anterior edge of the DA9 synaptic domain locally inactivates Rap2 GTPase to delineate the synaptic tiling border in DA9.

## *mig-15*(*TNIK*) regulates synaptic tiling

In mammals, TNIK (Traf2 and Nck1-interacting kinase) acts with Rap2 to regulate neurite extension, AMPA receptor trafficking in hippocampal neurons and microvilli formation in intestinal cells

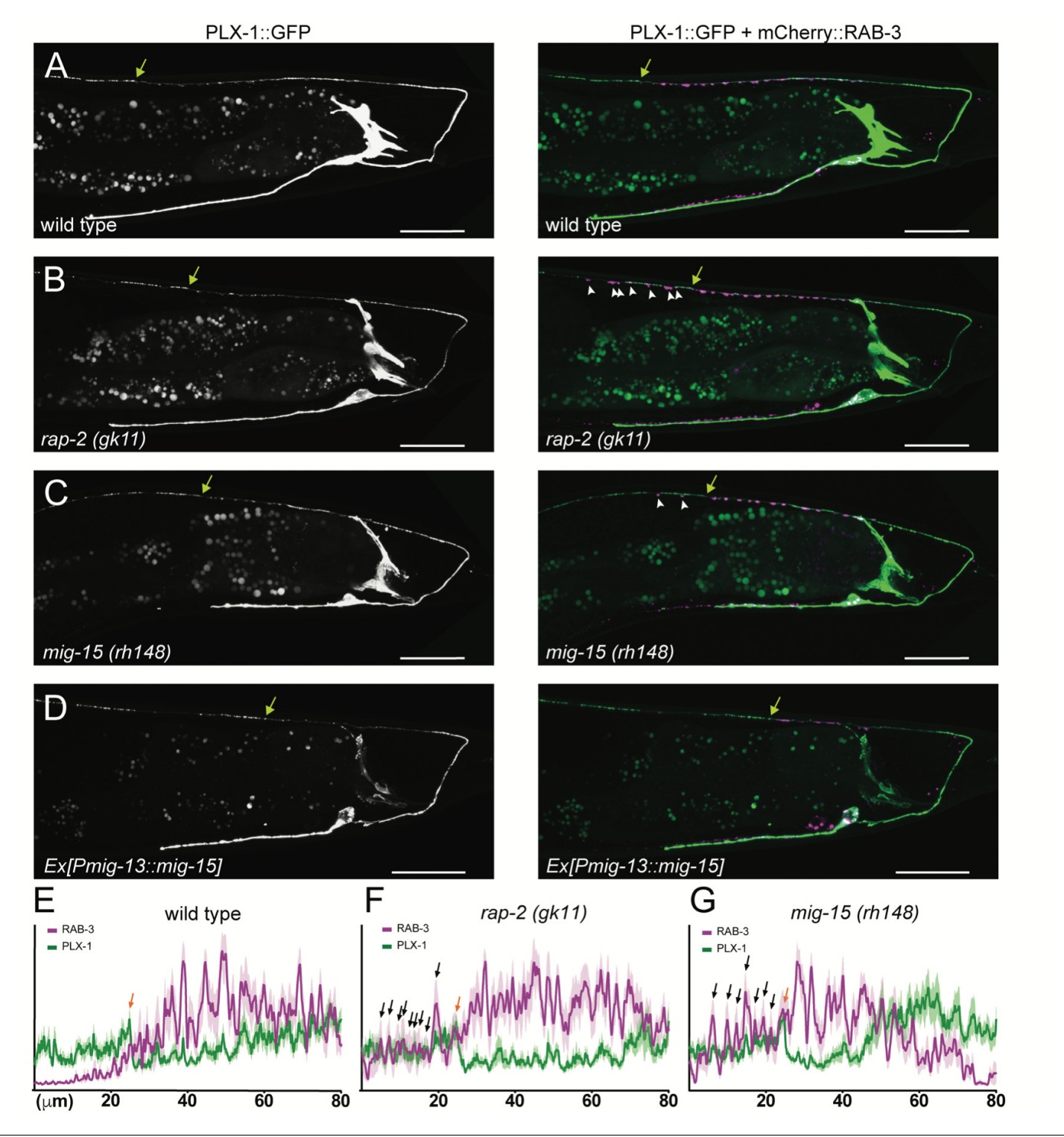

**Figure 3.** PLX-1::GFP localization is not affected in *rap-2* and *mig-15* mutants. (A–D) Representative image of PLX-1::GFP alone (top) and PLX-1::GFP with mCherry::RAB-3 (middle) labeled with *wyIs320* in DA9 of wildtype (A), *rap-2 (gk11)* (B) and *mig-15(rh148)* (C). Bracket indicates the PLX-1::GFP patch localized at the anterior edge of the DA9 synaptic domain. Arrowheads indicate ectopic synapses formed anterior to the PLX-1::GFP patch. Scale bars: 20 μm. (E–G) Quantification of the normalized mCherry::RAB-3 signal (magenta) and PLX-1::GFP signal (green) in the dorsal axon of DA9 in wildtype (E), *rap-2 (gk11)* (F) and *mig-15(rh148)* (G). Animal were aligned according to the PLX-1::GFP patch at the anterior edge of the DA9 synaptic domain (orange arrow). Light colors indicate SEM.

DOI: https://doi.org/10.7554/eLife.38801.008

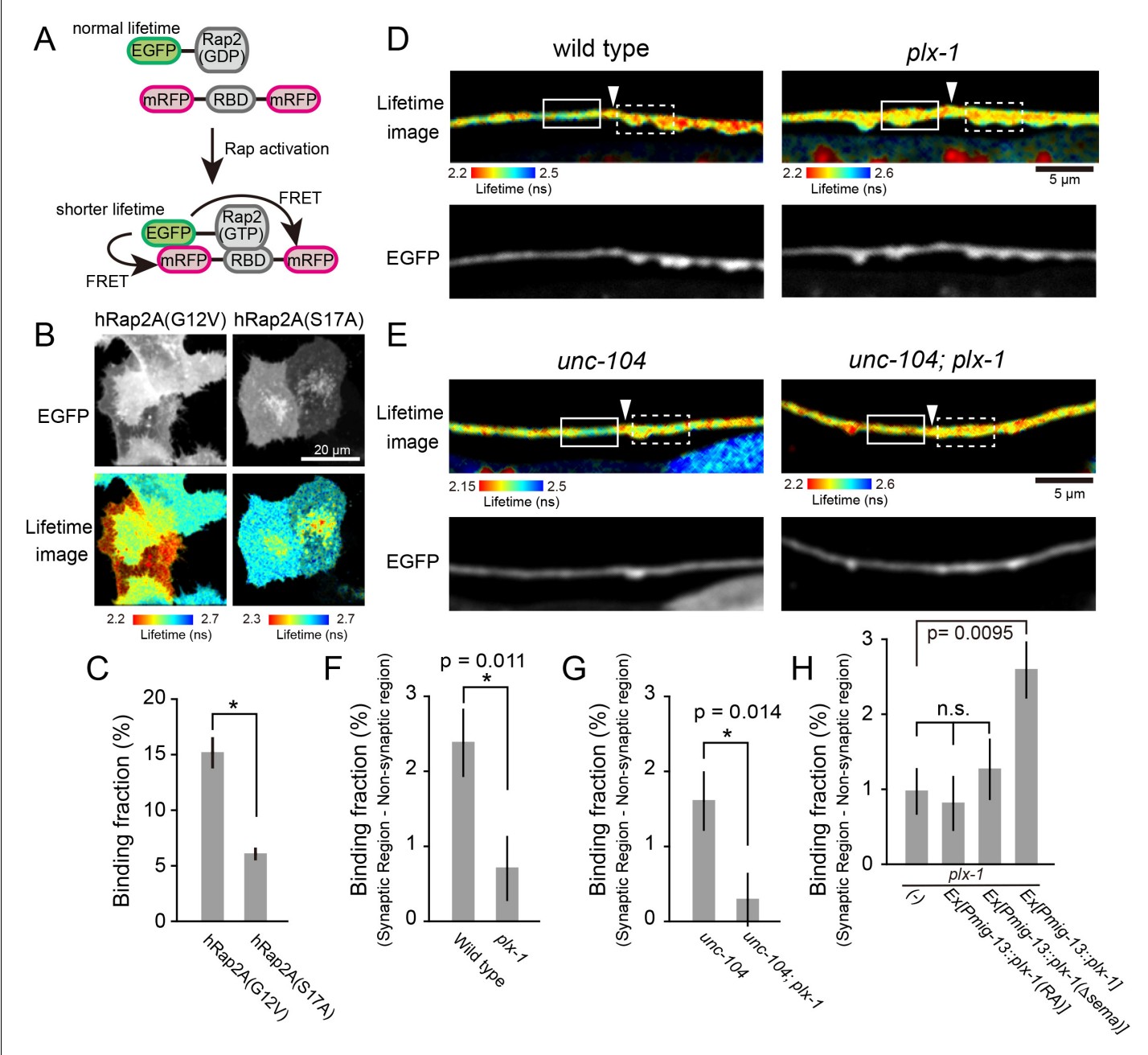

**Figure 4.** PLX-1 locally inhibits Rap2 activity in DA9. (**A**) Schematic of Rap2 FRET sensor system. Binding of mRFP- RalGDS(RBD)-mRFP to EGFP-Rap2 induces FRET from EGFP to mRFP, leading to decreased EGFP fluorescence lifetime. (**B**) Representative fluorescence (top) and fluorescence lifetime images (bottom) of HeLa cells expressing Rap2 FRET sensor (mRFP-RalGDS(RBD)-mRFP) with either constitutively GTP(G12V)- or GDP(S17A)- forms of human Rap2A. (**C**) Quantification of the binding fraction of Rap2 mutants in HeLa cells. Binding fractions measured from fluorescence decay curves of individual cells (G12V, n = 25; S17A, n = 25), as described previously (*Murakoshi et al., 2011*). Data shown as mean ± SEM. Asterisks denote statistical significance (p<0.05, student's t-test). (**D**) Representative images of fluorescence lifetime (top) and fluorescence (bottom) of *mizIs19* in wild type (left) and *plx-1* mutants (right). White arrowheads indicate the position of the putative synaptic tiling border, as judged by the slight dorsal shift of the DA9 axon, as reported previously (*Mizumoto and Shen, 2013a*). (**E**) Representative images of fluorescence lifetime (top) and fluorescence (bottom) of *mizIs19* in *unc-104* (left) and *unc-104; plx-1* double mutants (right). (**F**) Quantification of the difference in binding fraction of GTP-Rap2a and RalGDS (RBD) between synaptic region (dotted boxes in **D**) and anterior asynaptic region (solid boxes in **D**). The binding fraction measured in synaptic region (dotted rectangle) was subtracted by that in non-synaptic region (solid rectangle). Five micrometers along the axon line from the synaptic tiling border were used for quantification. Data presented as mean ± SEM (wild type, n = 22; *plx-1*, n = 17). (**G**) Quantification of the difference in binding fraction of GTP-Rap2A and RalGDS(RBD) between synaptic region (dotted boxes in **E**) and anterior asynaptic region (solid boxes in **E**). The binding fraction measured in synaptic region (dotted rectangle) subtracted from that in non-synaptic region (solid rectangle). Five micrometers along the axon line from

*Figure 4 continued on next page*

Figure 4 continued

the synaptic tiling border were used for quantification. Data are presented as mean ± SEM (*unc-104*, n = 21; *unc-104; plx-1*, n = 23). (**H**) Quantification of the difference in binding fraction of GTP-Rap2a and RalGDS(RBD) between synaptic region and anterior asynaptic region in *plx-1* mutants and *plx-1* mutants expressing rescuing constructs. Five micrometers along the axon line from the synaptic tiling border were used for quantification. Data are presented as mean ± SEM (*no array*, n = 35; *Ex[Pmig-13::plx-1(RA)]*, n = 39; *Ex[Pmig-13::plx-1(Δsema)]*, n = 41; *Ex[Pmig-13::plx-1]*, n = 51).
DOI: https://doi.org/10.7554/eLife.38801.009

(*Hussain et al., 2010*; *Kawabe et al., 2010*; *Gloerich et al., 2012*). In *C. elegans*, *mig-15* is the sole ortholog of mammalian TNIK and its paralog MINK1 (Misshapen-like kinase 1), which also is an effector of Rap GTPase (*Nonaka et al., 2008* ). *mig-15* can regulate various cellular processes, such as axon guidance and cell migration (*Chapman et al., 2008*; *Poinat et al., 2002*; *Shakir et al., 2006*; *Teulière et al., 2011*). We found that *mig-15(rh148)* hypomorphic mutants showed a severe synaptic tiling defect (*Figure 5A and D*). Similar to *plx-1* and *rap-2* mutants, the synaptic tiling defect followed the anterior expansion of the DA9 synaptic domain and the posterior expansion of the DA8 synaptic domain (*Figure 5E and F*). All RAB-3 puncta in *mig-15(rh148)* mutants co-localized with active zone markers, CLA-1 and UNC-10 (*Figure 1—figure supplement 3*), suggesting that these RAB-3 puncta represent bona fide synapses. We observed axon guidance defects (23%, n = 100) or ectopic branch formation (56%, n = 100) in DA9 of *mig-15* mutant animals (*Figure 5—figure supplement 1*). These were excluded from our analysis of synaptic tiling phenotypes. While only half of the *mig-15* mutant animals showed axon guidance defects or ectopic branch formation, the synaptic tiling defect of *mig-15* mutants was almost fully penetrant (*Figure 5D*). We did not observe significant synaptic tiling defects in *cdh-4(rh310)* mutants (DA8/DA9 overlap: 4.6 ± 1.02 μm, n = 21), which can exhibit an axon defasciculation phenotype in the dorsal nerve cord neurons (*Schmitz et al., 2008*). These data suggest that the synaptic tiling defect in the *mig-15* mutants is not a secondary effect of axon outgrowth and guidance.

The other two nonsense alleles (*rh326: Q439Stop, rh80: W898Stop*) also showed identical synaptic tiling defects as *mig-15(rh148)* (*Figure 5—figure supplement 2*). *mig-15(rh80)* has a nonsense mutation within the highly conserved CNH (citron/NIK homology) domain, which is required to interact with Rap2 in both mammals and *C. elegans* (*Taira et al., 2004*). This suggests a physical interaction between RAP-2 and MIG-15 for synaptic tiling. *plx-1* or *rap-2* mutants did not enhance the synaptic tiling defect in *mig-15* mutants (*Figure 5B–F*). This result is consistent with our hypothesis that *mig-15* acts in the same genetic pathway as *plx-1* and *rap-2*. The PLX-1::GFP patch at the putative synaptic tiling border was unaffected in *mig-15* mutants, even though the position of the PLX-1::GFP patch has shifted slightly posteriorly compared with wild type (*Figure 3C and G*). Taken together, these results suggest that *mig-15* acts downstream of *plx-1* to regulate synaptic tiling.

Interestingly, the degree of overlap between DA8 and DA9 synaptic domains was even larger in *mig-15* mutants than those observed in *plx-1* and *rap-2* mutants (compare *Figures 1G* and *5D*), suggesting that *mig-15* also acts downstream of additional signaling pathways (see discussion).

## *mig-15* functions in DA neurons

We then determined in which cells *mig-15* functions by conducting tissue specific rescue experiments. Since several *mig-15* isoforms (wormbase and data not shown) exist, we used the *mig-15* genomic sequence for the rescue experiments. Expression of *mig-15* under the DA neuron specific promoter (*Punc-4c*) strongly rescued the synaptic tiling defect of *mig-15(rh148)* mutants (*Figure 6A and F*), consistent with our hypothesis that *mig-15* acts in the same genetic pathway as *plx-1* and *rap-2*.

Expression of *mig-15* in both DA8 and DA9 rescued both posterior expansion of the DA8 synaptic domain and anterior expansion of the DA9 synaptic domain (*Figure 6G and H*). DA9 specific expression of *mig-15* under the *mig-13* promoter rescued anterior expansion of the DA9 synaptic domain, suggesting that *mig-15* functions cell autonomously in DA9 (*Figure 6B and G*). We observed that P*mig-13::mig-15* weakly rescued the posterior expansion of the DA8 synaptic domain (*Figure 6H*). This is likely due to the leaky expression of *mig-15* in DA8, as the *mig-15* genomic fragment without promoter showed slight rescue of the synaptic tiling defect in *mig-15(rh148)* mutants (*Figure 6F*). Kinase dead TNIK mutants act as a dominant-negative (*Mahmoudi et al., 2009*). In DA

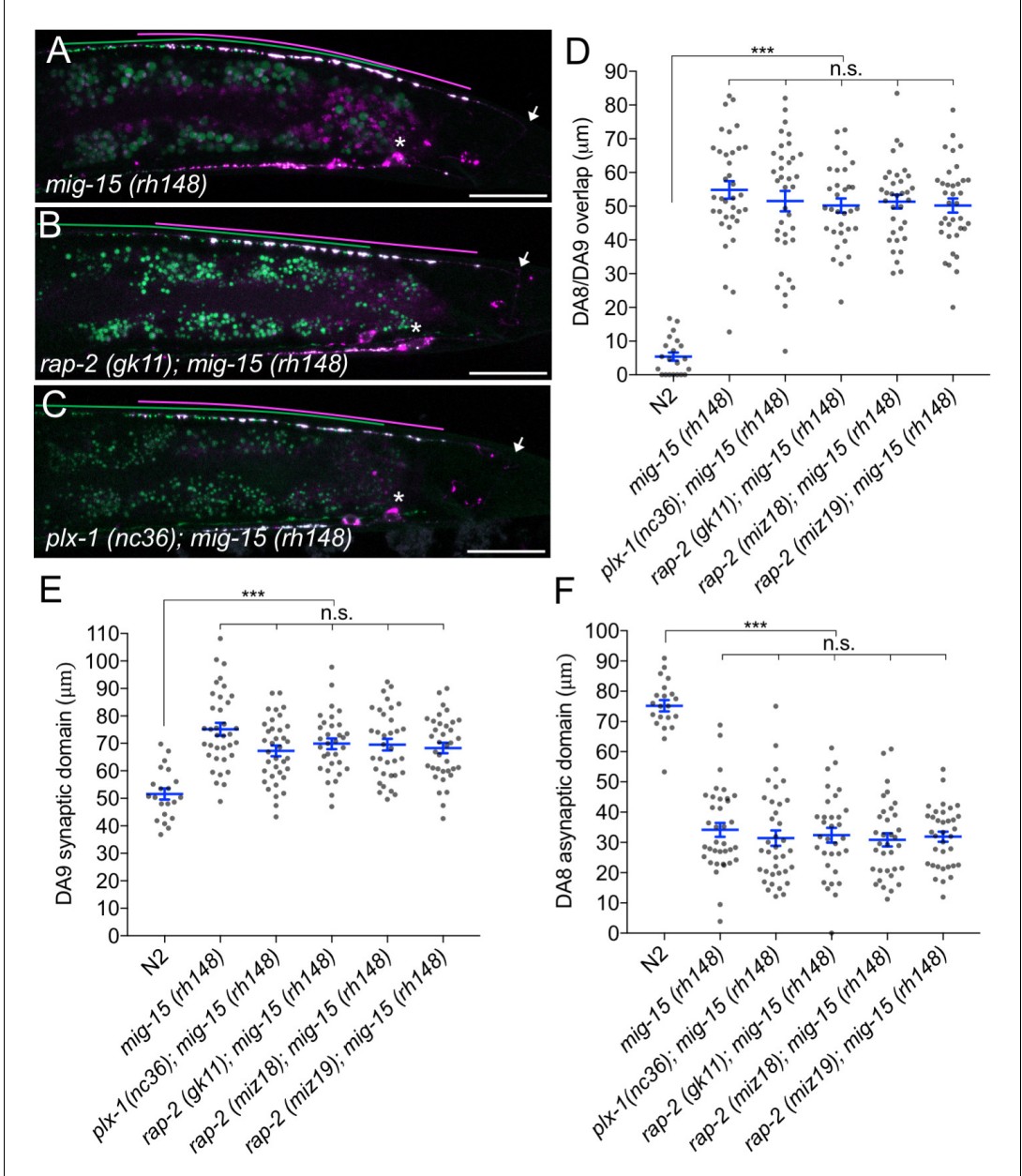

**Figure 5.** *mig-15(TNIK)* mutants show a severe synaptic tiling defect. (A–C) Representative images of synaptic tiling marker (*wyIs446*) in *mig-15(rh148)* (A), *rap-2(gk11); mig-15(rh148)* (B) and *plx-1(nc36); mig-15(rh148)* (C) mutants. Synaptic domains of DA8 and DA9 are highlighted with green and magenta lines, respectively. Asterisks: DA9 cell body. Arrows: dorsal commissure of DA9. Scale bars: 20 μm. (D–F) Quantification of overlap between DA8 and DA9 synaptic domains (D), DA9 synaptic domain (E) and DA8 asynaptic domain (F) in respective mutant backgrounds. Each dot represents measurement from a single animal. Blue bars indicate mean ± SEM. n.s.: not significant; ***p<0.001.
DOI: https://doi.org/10.7554/eLife.38801.010

The following figure supplements are available for figure 5:

**Figure supplement 1.** Morphological defects of DA9 axon in *mig-15(rh148)* mutants.
DOI: https://doi.org/10.7554/eLife.38801.011

**Figure supplement 2.** Synaptic tiling defect in *mig-15* mutants.
DOI: https://doi.org/10.7554/eLife.38801.012

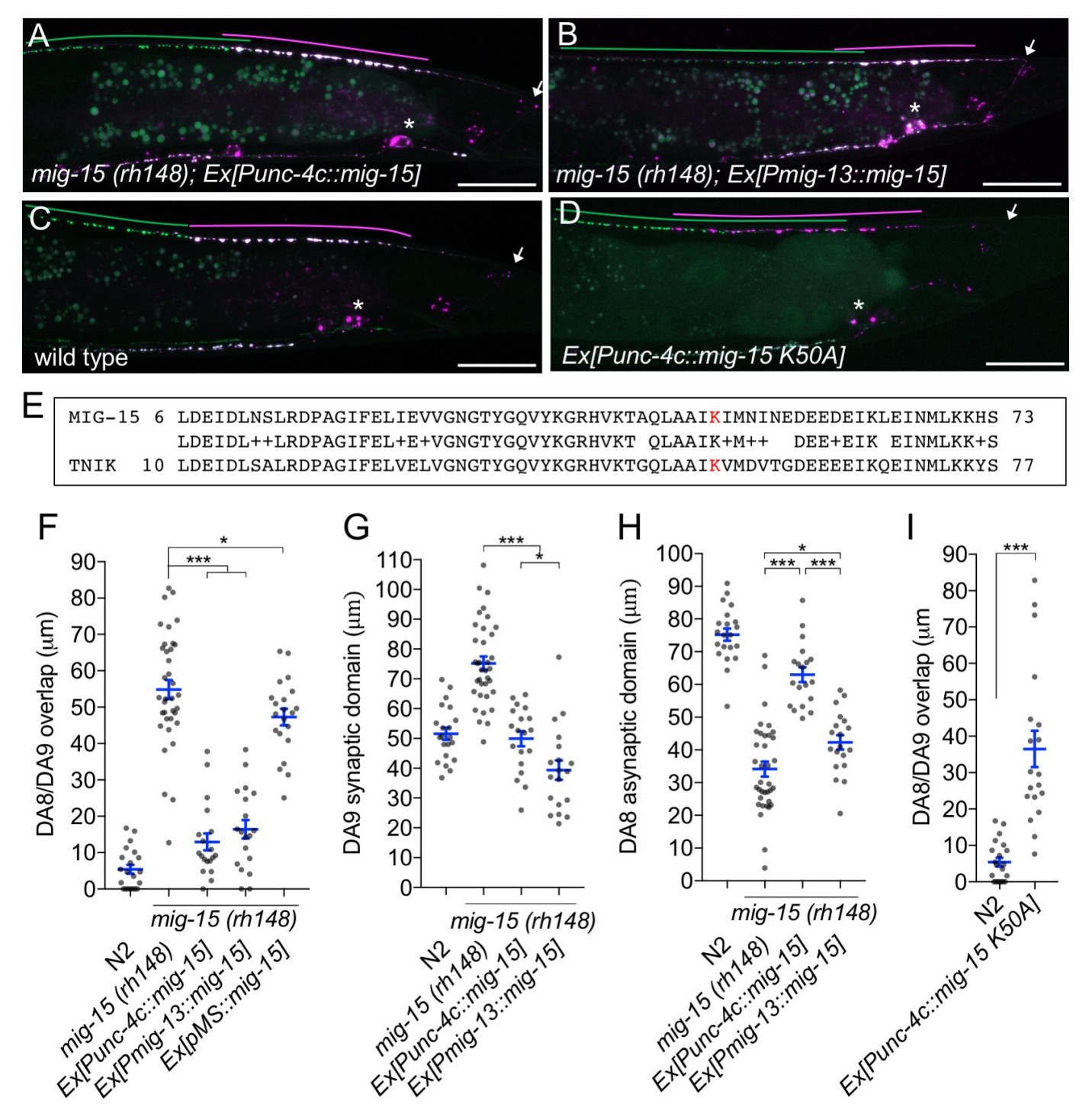

**Figure 6.** *mig-15* functions in DA neurons. (A–D) Representative images of synaptic tiling marker (*wyIs446*) in the following backgrounds: *mig-15(rh148); Ex[Punc-4c::mig-15]* (A), *mig-15(rh148); Ex[Pmig-13::mig-15]* (B), wild type (C) and wild type animals expressing dominant-negative *mig-15(K50A)* in DA neurons (D). Synaptic domains of DA8 and DA9 are highlighted with green and magenta lines, respectively. Asterisks: DA9 cell body. Arrows: dorsal commissure of DA9. Scale bars: 20 μm. (E) Amino acid alignment of amino-terminal region of MIG-15 and TNIK. A kinase-dead mutation in TNIK (K54A) and corresponding mutation in MIG-15 (K50A) are highlighted in red. (F–I) Quantification of overlap between DA8 and DA9 synaptic domains (F and I), DA9 synaptic domain (G), DA8 asynaptic domain (H) in respective mutant backgrounds. Each dot represents measurements from a single animal. Blue bars indicate mean ± SEM. n.s.: not significant; ***p<0.001; *p<0.05.

DOI: https://doi.org/10.7554/eLife.38801.013

neurons, expression of mutant *mig-15(kd)*, which carries the same mutation at the corresponding amino acid of the dominant-negative TNIK (*Figure 6E*), in DA neurons caused a severe synaptic tiling defect (*Figure 6C, D and I*). Based on these results, we conclude that *mig-15* functions cell autonomously in DA neurons.

## *mig-15* inhibits synapse formation

We observed that DA9-specific expression of *mig-15* under the *mig-13* promoter in *mig-15* mutants often exhibited a shorter synaptic domain compared to wild type (*Figure 6B and G*). So, we speculated that an excess amount of *mig-15* inhibits synapse formation. We tested the effect of *mig-15* overexpression in the wild type background. Strikingly, DA9-specific *mig-15* overexpression in wild type (*mig-15(OE)*) significantly reduced synapse number compared to wild type (*Figure 7A, C and D* and *Figure 7—figure supplement 1*). This reduction occurred without affecting the overall morphology of the DA9 neuron (*Figure 5—figure supplement 1*). Conversely, DA9 synapse number was significantly increased in the *mig-15(rh148)* mutants (*Figure 7B and D* and *Figure 7—figure supplement 1*). *mig-15* overexpression also significantly reduced synapse number in DD-type GABAergic motor neurons (*Figure 7—figure supplement 2*). These results indicate that *mig-15* is a negative regulator of synapse formation. Further, pan-neuronal expression of *mig-15* under the *rab-3* promoter caused severe uncoordinated locomotion in wildtype animals (*Figure 7—figure supplement 3*). These locomotor defects occurred concomitant with significantly reduced GFP::RAB-3 intensity in the dorsal nerve cord in *mig-15* over-expressing animals and without causing significant axon guidance defects (*Figure 7—figure supplement 3*). Taken together, these data indicate that reduced synapse number by *mig-15* overexpression disrupted proper functioning of the motor circuit. Importantly, we observed no significant increase in synapse numbers in *plx-1* or *rap-2* mutants (*Figure 7—figure supplement 1*), suggesting that the role of *mig-15* in negatively regulating synapse number is independent of its role in PLX-1/RAP-2 -mediated synaptic tiling (*Figure 8G*).

Rap GTPase and TNIK are well-known actin cytoskeleton regulators (*Lin et al., 2010*, *2008*; *Taira et al., 2004*). Previous studies demonstrated that presynaptic development requires ARP2/3-dependent branched F-actin (*Chia et al., 2012*, *2014*). Branched F-actin visualized by GFP::ut-CH (utrophin calponin homology domain) is enriched within the DA9 synaptic domain (*Figure 7E*) (*Chia et al., 2012*; *Mizumoto and Shen, 2013a*). We predicted that *mig-15* negatively regulates synapse formation by re-organizing branched F-actin at the anterior edge of the synaptic domain. Consistently, we observed longer synaptic F-actin distribution in *rap-2(gk11)* and *mig-15(rh148)* mutants (*Figure 7F, G and I*). While GFP::ut-CH was observed in the posterior asynaptic axonal region or in the dendrite of DA9, synapse formation is likely inhibited by Wnt and Netrin signaling as previously reported (*Klassen and Shen, 2007*; *Poon et al., 2008*). Conversely, overexpression of *mig-15* in DA9 significantly decreased the length of synaptic F-actin (*Figure 7H and I*). Overexpression of *mig-15* also appeared to decrease the overall amount of synaptic F-actin (*Figure 7H*). This result suggests that *mig-15* inhibits synapse formation by negatively regulating the formation of synaptic F-actin.

## *plx-1* and *rap-2* coordinate synaptic tiling border positioning and synapse number

*mig-15(OE)* reduced the number of synapses in DA9. As a result, the length of the DA9 synaptic domain was significantly reduced in *mig-15(OE)* animals than in wild type (*Figure 8A and E*). Yet, synaptic tiling is maintained without a significant gap between DA8 and DA9 synaptic domains in *mig-15(OE)* animals, suggesting that the position of synaptic tiling border shifted posteriorly in *mig-15(OE)* animals. Indeed, the length of the posterior asynaptic domain of DA8 was significantly shorter in *mig-15(OE)* animals, indicating that the DA8 synaptic domain expanded posteriorly (*Figure 8D*). The PLX-1::GFP patch at the anterior edge of the DA9 synaptic domain also shifted posteriorly in *mig-15(OE)* animals with reduced synapse number (*Figure 3D*). These results strongly suggest that the PLX-1/RAP-2 signaling pathway may specify the position of the synaptic tiling border according to the available number of synapses in each DA neuron (*Figure 8G*). We propose that synaptic tiling is a mechanism to maintain a uniform distribution of synapses from one class of motor neuron in the nerve cord. Consistently, we found that DA8 synaptic domain did not shift posteriorly when *mig-15* was overexpressed in DA9 of the synaptic tiling mutants, *plx-1* or *rap-2* (*Figure 8B–E*).

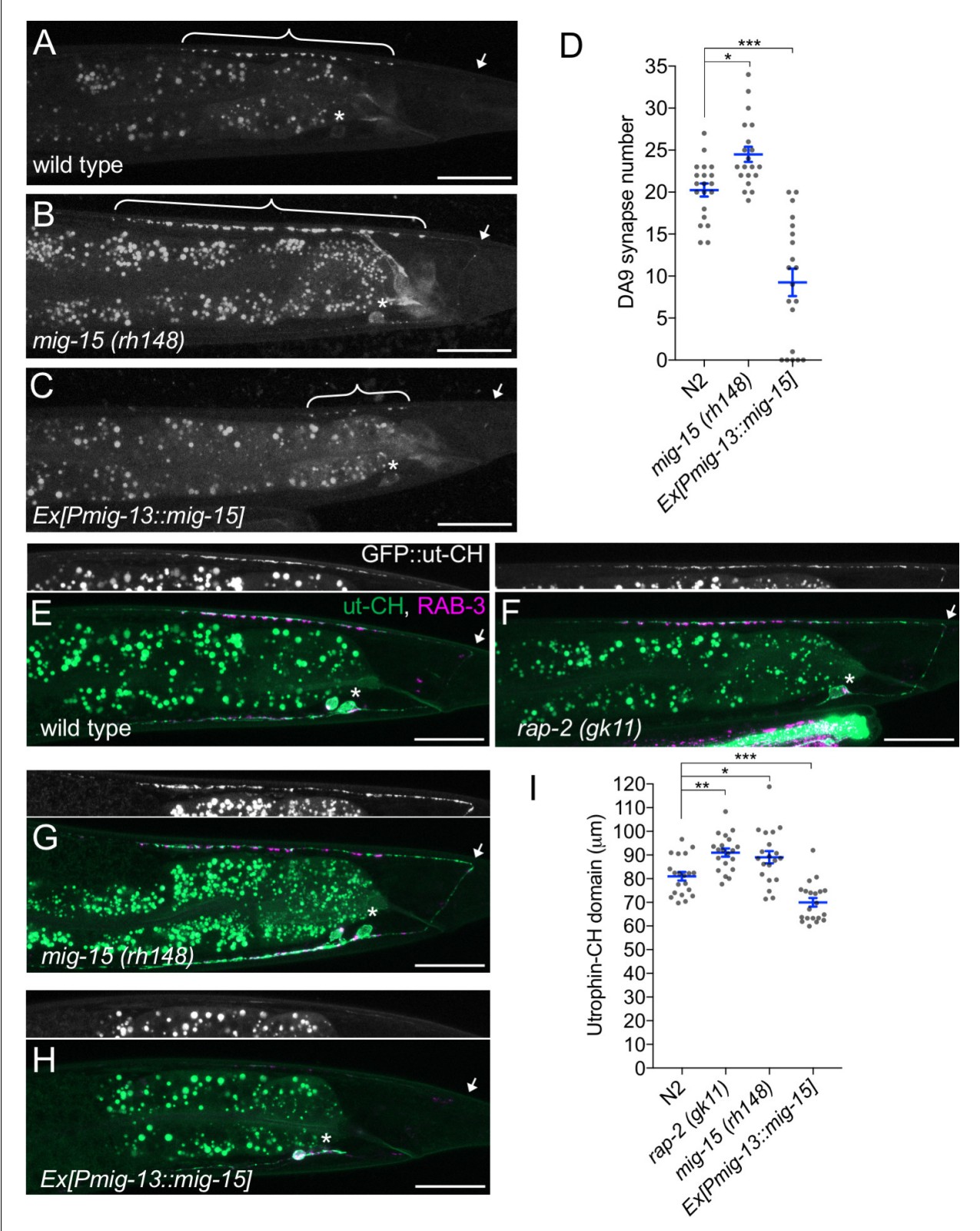

**Figure 7.** *mig-15* negatively regulates synapse number. (A–C) Representative images of DA9 presynaptic marker, GFP::RAB-3 (*wyIs85*), in wild type (A), *mig-15(rh148)* (B) and *mig-15* overexpressing animals (C). Brackets represent DA9 synaptic domain. Asterisks: DA9 cell body. Arrows: dorsal commissure of DA9. Scale bars: 20 μm. (D) Quantification of DA9 synapse number. Each dot represents measurements from a single animal. Blue bars indicate mean ± SEM. n.s.: not significant; ***p<0.001; *p<0.05. (E–H) Representative images of synaptic branched F-actin labeled with GFP::ut-CH (*wyIs329*) in
*Figure 7 continued on next page*

*Figure 7 continued*

wild type (E), *rap-2(gk11)* (F), *mig-15(rh148)* (G) and *mig-15* overexpressing animals (H). (I) Quantification of the length of GFP::ut-CH. Distance from the dorsal commissure to the most anterior and brightest GFP spot was measured. Blue bars indicate mean ± SEM. n.s.: not significant; \*\*\*p<0.001; \*\*p<0.01, \*p<0.05.

DOI: https://doi.org/10.7554/eLife.38801.014

The following figure supplements are available for figure 7:

**Figure supplement 1.** *mig-15* is a negative regulator of synapse formation.

DOI: https://doi.org/10.7554/eLife.38801.015

**Figure supplement 2.** Overexpression of *mig-15* reduces synapse numbers in GABAergic motor neurons.

DOI: https://doi.org/10.7554/eLife.38801.016

**Figure supplement 3.** Reduction of synapses by *mig-15* overexpression caused severe locomotion defects.

DOI: https://doi.org/10.7554/eLife.38801.017

This result suggests that DA8 no longer senses the reduction of DA9 synapse number in the synaptic tiling mutants. Synapse number was not different between *mig-15(OE)* and in *rap-2(gk11); mig-15 (OE)* animals (*Figure 7—figure supplement 1*), suggesting that *mig-15* is not dependent on the Plexin/Rap2 signaling pathway to inhibit synapse number (*Figure 8G*).

In summary, we demonstrate that synaptic tiling maintains a uniform distribution of synapses from one class of motor neurons along the nerve cord. Further, our results indicate *plx-1* and *rap-2* play critical roles in this process by coordinating the position of the synaptic tiling border.

## Discussion

While much is known about the morphogenic triggers for axon guidance and patterned synapse formation, the downstream sequelae of these intracellular effectors have remained unclear. We discovered the role of Rap2 GTPase and TNIK in synapse pattern formation in *C. elegans*. Since Sema/Plexin signaling processes to inhibit synapse formation are well conserved across species, we propose that Sema/Plexin in mammals also utilize Rap2 and TNIK to regulate synapse patterning.

### Cell autonomous and non-autonomous functions of Sema/Plexin signaling components

Previously we showed that both *smp-1* and *plx-1* are necessary and sufficient in DA9, which suggests that *smp-1* and *plx-1* act cell-autonomously in DA9 and non-autonomously in DA8 to regulate synaptic tiling. We proposed that Sema/PLX-1 in DA9 send a retrograde signal to DA8 through an unidentified signaling molecule (X) to induce the synaptic tiling pattern in DA8 (*Mizumoto and Shen, 2013a*). However, we found that both *rap-2* and *mig-15* act cell autonomously, since our DA9-specific rescue experiment only rescued the DA9 phenotype, but not the DA8 phenotype. This conclusion is further supported since the synaptic tiling defects of these mutants were fully rescued when both neurons express functional *rap-2* cDNA or *mig-15* genomic DNA. We propose that each neuron utilizes a different set of cell surface proteins but share common intracellular mechanisms to specify synapse patterning. Diverse signaling and cell adhesion molecules, such as atrial natriuretic peptide receptor (NPR) and GPCRs, regulate Rap activity (*Gloerich and Bos, 2011*; *Birukova et al., 2008*; *Weissman et al., 2004*). Screening for these potential Rap regulators should identify novel molecules that interact with Sema/Plexin and act in DA9.

### Cycling of Rap GTPase activity in synaptic tiling

We showed that proper synapse patterning requires both GDP- and GTP-forms of RAP-2. Considering that PLX-1 regulates the spatial distribution of RAP-2 activity and *mig-15* acts genetically downstream of *plx-1* in synaptic tiling, RAP-2 may also locally regulate MIG-15 (TNIK). While we did not observe a specific subcellular localization of GFP-MIG-15 in DA9 (data not shown), PLX-1 and RAP-2 may instead regulate MIG-15 activity rather than its spatial localization. Further biochemical characterization of MIG-15 regulation by GTP-RAP-2 or GDP-RAP-2 will elucidate the exact functions of RAP-2 in synapse patterning.

Small GTPase activity is regulated by GAP and GEF (Guanine nucleotide exchange factor) proteins. Yet, we did not observe significant synaptic tiling defects in mutants of putative RAP-2 GEFs,

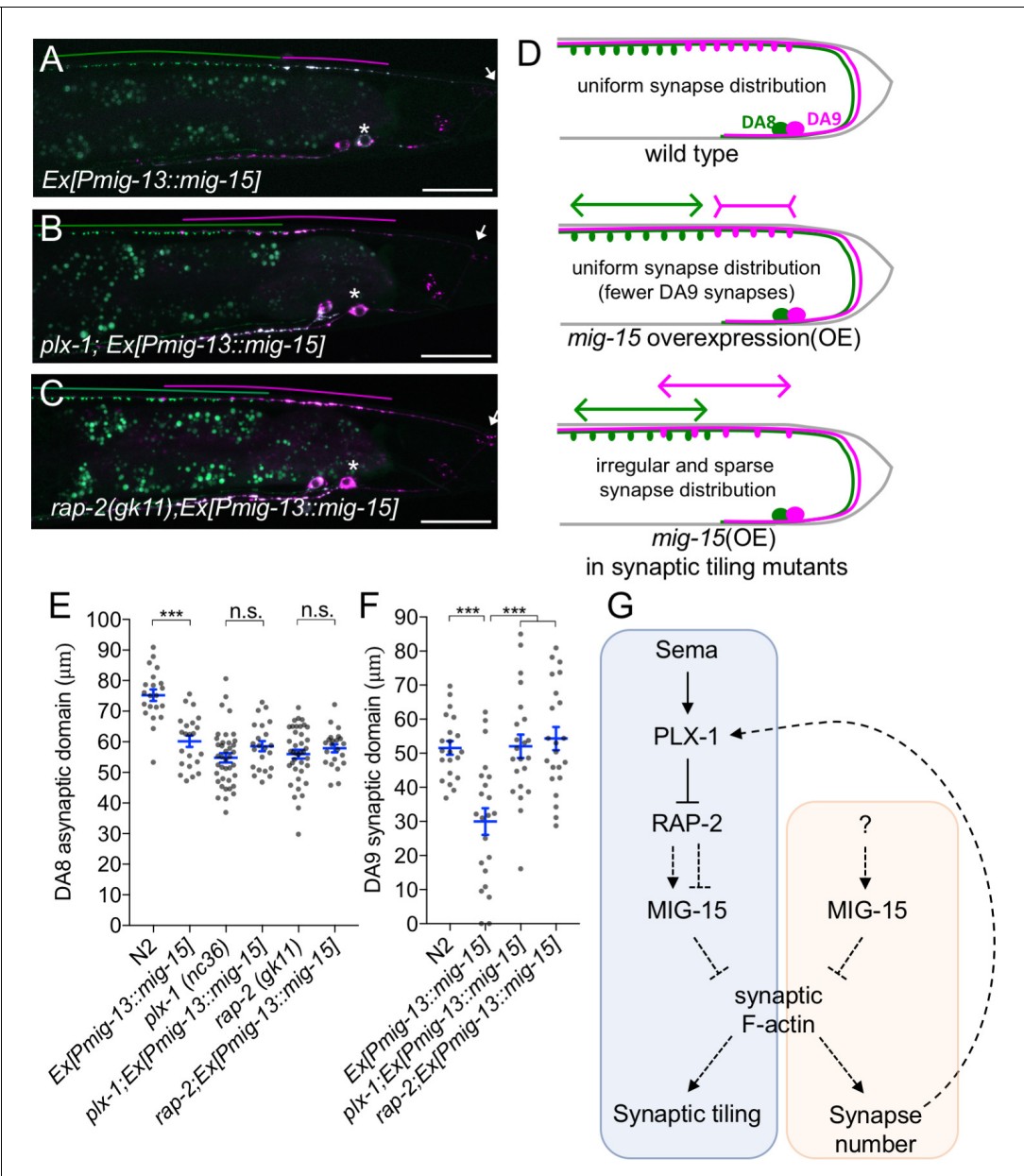

**Figure 8.** PLX-1/RAP-2 signaling coordinates synapse number and synaptic tiling border. (**A–C**) Representative images of synaptic tiling marker (*wyIs446*) overexpressing *mig-15* in DA9 from wild type (**A**), *plx-1(nc36)* (**B**) and *rap-2(gk11)* (**C**). Synaptic domains of DA8 and DA9 highlighted with green and magenta lines, respectively. Asterisks: DA9 cell body. Arrows: dorsal commissure of DA9. Scale bars: 20 μm. (**D**) Schematic illustration of synapse distribution in wild type (left), animals overexpressing *mig-15* in DA9 (middle) and synaptic tiling mutants overexpressing *mig-15* in DA9 (right). Arrows indicate the synaptic tiling border. Colored arrows represent expanded or shortened synaptic domain from DA8 (green) and DA9 (magenta). (**E**) Quantification of the DA8 asynaptic domain. (**F**) Quantification of the DA9 synaptic domain. Each dot represents measurements from a single animal. Blue bars indicate mean ± SEM. n.s.: not significant; ***p<0.001. (**G**) A model of synaptic tiling. PLX-1/RAP-2 signaling controls synaptic tiling via MIG-15, while MIG-15 also plays crucial roles in inhibiting the number of synapses. Solid lines indicate putative direct regulations, dotted lines represent indirect or unknown mode of regulations.

DOI: https://doi.org/10.7554/eLife.38801.018

which include *pxf-1(RAPGEF2/6)* and *epac-1(RAPGEF3/4/5)* (data not shown) (*Frische et al., 2007*; *Pellis-van Berkel et al., 2005*). We speculate that multiple RapGEFs act redundantly to activate RAP-2 in synaptic tiling.

## *mig-15(TNIK)* may integrate multiple inhibitory cues during synapse formation

*mig-15* mutants show a greater degree of overlap between DA8 and DA9 synaptic domains than *plx-1* or *rap-2* mutants. This effect partially occurs from excess synaptogenesis in the posterior asynaptic domain of both DA8 and DA9 neurons. Previously, we demonstrated that Wnt morphogens and their receptors, Frizzled, instruct synaptic topographic patterning by locally inhibiting synapse formation. Indeed, synaptic tiling defects in *mig-15* mutants was somewhat similar to the combined effect of *plx-1* and *wnt* mutants (*Mizumoto and Shen, 2013b*). TNIK can act as a positive regulator of the canonical Wnt signaling pathway in colorectal cancer cells (*Mahmoudi et al., 2009*; *Shitashige et al., 2010*). While we do not know whether the canonical Wnt signaling pathway contributes to local inhibition of synapse formation, we propose that TNIK integrates multiple signaling pathways for precise synapse pattern formation.

In addition to its role in synapse pattern formation, our data indicate that *mig-15* also plays a role as a negative regulator of synapse number. Since neither *plx-1* nor *rap-2* mutants showed a significant increase in synapse number in DA9, *mig-15* may inhibit synapse formation in a different signaling pathway (*Figure 8G*). As we observed a global reduction of synaptic actin staining in animals over-expressing *mig-15,* we propose that *mig-15* controls synapse number by regulating synaptic F-actin.

The exact mechanisms of synaptic actin regulation by TNIK remain undetermined. TNIK could activate the JNK kinase pathway (*Taira et al., 2004*; *Fu et al., 1999*). The MIG-15/JNK-1 signaling pathway inhibits axonal branch formation in s *C. elegans* ensory neurons (*Crawley et al., 2017*). In contrast to these well-established roles of MIG-15/TNIK as an activator of the JNK pathway, we did not observe any synaptic tiling defect nor change in synapse number in *jnk-1* mutant animals (*Figure 5—figure supplement 2* and data not shown). Our results suggest *mig-15* does not inhibit synapse formation through the JNK pathway.

Due to the pleiotropic phenotype of the *mig-15* mutants, our genetic and phenotypic analyses of *mig-15* did not exquisitely reveal the mechanistic relationship between PLX-1 and MIG-15 in synaptic tiling regulation. Further biochemical studies of MIG-15 regulation by Plexin/Rap2 in synaptic tiling will elucidate the molecular mechanisms that underlie the role of MIG-15/TNIK in synapse pattern formation.

## Plexin signaling and diseases

Aberrant neuronal wiring underlies many neurological disorders. Not surprisingly, Semaphorin and Plexin genes are associated with various neurodevelopmental disorders and intellectual disabilities, including autism spectrum disorders (ASD) and schizophrenia (*Mah et al., 2006*; *Gene Discovery Project of Johns Hopkins & the Autism Consortium et al., 2009*). For example, PLXNB1, SEMA3A, SEMA4D and SEMA6C are significantly upregulated in the prefrontal cortices of schizophrenic patients (*Eastwood et al., 2003*; *Gilabert-Juan et al., 2015*). On the other hand, non-synonymous variations in the Sema3D gene had a significant protective effect against developing schizophrenia (*Fujii et al., 2011*). More recent work showed that loss of Sema5A/PlexA2 signaling induces excess excitatory synapse formation in granule cells, which caused ASD-like behavioural defects in mice (*Duan et al., 2014*).

Similar to Sema/Plexin signaling, TNIK is also associated with various neurological disorders, including schizophrenia and intellectual disabilities (*Anazi et al., 2016*; *Potkin et al., 2010*). TNIK can also physically bind and act with DISC1 (Disrupted in Schizophrenia 1) to regulate synaptic composition (*Wang et al., 2011*). So, we propose that the Sema/Plexin/Rap2/TNIK signaling pathway plays a critical role to precisely define synaptic connections and its disruption may induce serious neurological disorders.

Interestingly, SNPs in Plexin genes are also associated with extremely high IQ (*Spain et al., 2016*). Recent work suggests that loss of PlexinA1 confers better motor control in rodents due to increased synaptic connectivity in the corticospinal cord (*Gu et al., 2017*). Further studies on the Plexin/Rap2/TNIK signaling pathway in synapse map formation, as presented here, will likely reveal the genetic basis of these disorders and conditions.

# Materials and methods

**Key resources table**

| Reagent type (species) or resource | Designation | Source or reference | Identifiers | Additional information |
|---|---|---|---|---|
| Gene (*C. elegans*) | rap-2 | NA | C25D7.7 | |
| Gene (*C. elegans*) | plx-1 | NA | Y55F3AL.1 | |
| Gene (*C. elegans*) | mig-15 | NA | ZC504.4 | |
| Strain, strain background (*C. elegans*) | rap-2(gk11) | *C. elegans* stock center (CGC) | VC14 | |
| Strain, strain background (*C. elegans*) | mig-15(rh148) | *C. elegans* stock center (CGC) | NJ490 | |
| Strain, strain background (*C. elegans*) | plx-1(nc36) | *C. elegans* stock center (CGC) | ST36 | |
| Strain, strain background (*C. elegans*) | rap-2(miz18) | This study | UJ401 | G12V mutant |
| Strain, strain background (*C. elegans*) | rap-2(miz19) | This study | UJ402 | S17A mutant |
| Strain, strain background (*C. elegans*) | wyIs446 | This study | TV14517 | synaptic tiling marker |
| Strain, strain background (*C. elegans*) | mizIs19 | This study | UJ397 | Rap2 FRET sensor strain |

## Strains

All *C. elegans* strains were derived from Bristol N2 and raised on OP50 *Escherichia coli*-seeded nematode growth medium (NGM) plates at 20C and maintained as described previously (*Brenner, 1974*). The following mutants were used in this study: *unc-104(e1265)II, plx-1(nc36)IV, rap-1(pk2082)IV, rap-3(gk3975)IV, jnk-1(gk7)IV rap-2(gk11)V, rap-2(miz16)V, rap-2(miz17)V, rap-2(miz18)V, rap-2(miz19)V. rap-2(miz20)V, mig-15(rh148)X, mig-15(rh80)X, mig-15(rh326)X.*

## CRISPR/Cas9 genome editing

*rap-2(miz16)V, rap-2(miz17)V, rap-2(miz18)V, rap-2(miz19)V. rap-2(miz20)V* were generated using Co-CRISPR method (*Kim et al., 2014*). *unc-22* or *dpy-10* co-CRISPR markers were used for selecting candidate animals (*Kim et al., 2014*; *Arribere et al., 2014*). Vectors for sgRNA and Cas9 were obtained from Addgene (Plasmid ID: 46169 and 46168, respectively) (*Friedland et al., 2013*). The *rap-2* guide RNA sequence (5' – gTAGTGGAGGTGTCGGAAAAT-3') was designed using MIT CRISPR design tool (crispr.mit.edu:8079) and inserted into sgRNA vector using Q5 Site-Directed Mutagenesis kit (NEB). Repair templates with either G12V (*miz17* and *miz18*) and S17A (*miz19* and *miz20*) mutation were generated by PCR with primer sets carrying corresponding mutations (see supplemental Experimental Procesures). Synonymous mutations were also introduced in the sgRNA recognition sequence to avoid Cas9 recruitment to the edited genome. PCR products were cloned into *Eco*RI site of the pBluescript SK(+) vector. Synthesized double-stranded DNA (GeneArt, Thermo Fisher) was used as a repair template for generating *rap-2(miz16)* mutant.

## Plasmid constructions

*C. elegans* expression clones were made in a derivative of pPD49.26 (A. Fire), the pSM vector (kind gift from S. McCarroll and C. I. Bargmann). Primer sets used in this study are listed in the Supplemental Experimental Procedures. The following constructs were used and transgenes were generated using standard microinjection method (*Mello et al., 1991*): wyIs446 (*Punc-4::2xGFP-rab-3; Pmig-13::mCherry-rab-3; Podr-1::RFP*), wyIs85 (*Pitr-1::GFP-rab-3; Podr-1::RFP*), wyIs442 (*Pflp-13::2xGFP-rab-3; Pplx-2::mCherry-rab-3; Podr-1::RFP*), wyIs320 (*Pitr-1::plx-1::GFP; Pmig-13::mCherry;;rab-3; Podr-1::GFP*), wyIs329 (*Pmig-13::GFP-ut-CH; Pmig-13::mCherry::rab-3; Podr-1::GFP*), wyIs524 (*Punc-4::2xGFP-rab-3; Pmig-13::mCherry-rab-3; Podr-1::RFP*), wyIs685 (*Pmig-13::*

*mCherry::rab-3; Pmig-13::GFPnovo2::cla-1; Podr-1::GFP) mizIs1 (Pitr-1::GFPnovo2-CAAX; Pvha-6:: zif-1; Pitr-1::mCherry::rab-3; Podr-1::GFP), mizIs19 (Pmig-13::eGFP::hRap2a; Pmig-13::mRFP-RalGDS (RBD)-mRFP; Podr-1::GFP), mizIs33 (Prab-3::mig-15; Podr-1::GFP), jsIs682 (Prab-3::GFP::rab-3; lin-15 (+)), wyEx5445 (Prap-1::GFP; Punc-4::myr-mCherry; Podr-1::RFP), wyEx5464 (Prap-2::GFP; Punc-4:: myr-mCherry; Podr-1::RFP), mizEx194 (Prap-3::GFP; Pmig-13::myr-mCherry; Podr-1::RFP), mizEx165 (Phlh-1::rap-2; Podr-1::GFP), mizEx164 (Punc-129::rap-2; Podr-1::GFP), mizEx174 (Punc-4c::rap-2; Podr-1::GFP), mizEx157 (Pmig-13::rap-2; Podr-1::GFP), mizEx156 (Punc-4c::hRap2a; Podr-1::GFP), mizEx177 (Punc-4c::rap-1; Podr-1::GFP), mizEx151 (Pmig-13::mig-15; Podr-1::GFP), mizEx147 (Punc-4c::mig-15; Podr-1::GFP), mizEx153 (ΔpSMmig-15; Podr-1::GFP), mizEx178 (Punc-4c::mig-15(K50A); Podr-1::GFP), mizEx173 (Punc-4::rap-2(G12V); Podr-1::GFP), mizEx179 (Pflp-13::mig-15; Podr-1:: GFP), mizEx170 (Pmig-13::mig-15; Podr-1::GFP), mizEx197 (Pmig-13::mig-15; Podr-1::GFP), mizEx210 (Pmig-13::mig-15; Podr-1::RFP), mizEx257 (Prab-3::GFP; Prab-3::mCherry::rab-3, Podr-1:: mScarlet::CAAX), mizEx272 (Podr-1::GFP; Pmig-13::unc-10::TdTomato); mizEx309 (Pmig-13::plx-1 (RA); Podr-1::RFP), mizEx312 (Pmig-13::plx-1(Δsema); Podr-1::RFP); mizEx314 (Pmig-13::plx-1; Podr-1::RFP).*

## Cloning of *rap-1*, *rap-2* and *mig-15*

cDNAs of *rap-1* and *rap-2* were obtained from cDNA library prepared from N2 RNA. Trizol (Invitrogen) was used to purify total RNA from N2, and the SuperScript III First-Strand Synthesis System for RT-PCR (Invitrogen) was used for the reverse-transcription. *mig-15* genomic DNA was amplified from the N2 genomic DNA purified using GeneJET *Genomic* DNA Purification Kit (Thermo Scientific). Phusion (NEB) or Q5 (NEB) DNA polymerases were used for all PCR reactions for amplifying cDNA and genomic DNA fragments. Amplified fragments were cloned into the *Asc*I and *Kpn*I sites of pSM vector using SLiCE method (*Motohashi, 2015*), Gibson assembly (*Gibson et al., 2009*) or T4 ligase (NEB). List of primers used in this study is available in the Supplemental Experimental Procedures.

## Confocal microscopy

Images of fluorescently tagged fusion proteins were captured in live *C. elegans* using a Zeiss LSM800 confocal microscope (Carl Zeiss, Germany). Worms were immobilized on 2% agarose pad using a mixture of 7.5 mM levamisole (Sigma-Aldrich) and 0.225M BDM (2,3-butanedione monoxime) (Sigma-Aldrich). Images were analyzed with Zen software (Carl Zeiss) or Image J (NIH, USA). Definition of each parameter is as follows (*Mizumoto and Shen, 2013a*): DA8/9 overlap: a distance between the most anterior DA9 synapse and the most posterior DA8 synapse, DA8 asynaptic domain: a distance from commissure to the most posterior DA8 synapse, DA9 synaptic domain: a distance between the most anterior and posterior DA9 synapses. Middle L4 (judged by the stereotyped shape of developing vulva) animals were used for quantification. Averages were taken from at least 20 samples. For GFP::Utrophin-CH, we measured the length from the posterior end of dorsal axon to the anterior end of GFP::Utrophin-CH domain. For each marker strain, the same imaging setting (laser power, gain pinhole) and image processing were used for comparing different genotypes.

## Two-photon FLIM-FRET experiment

Expression vector for cultured cells (pCI-eGFP-hRap2a, pCI-eGFP-RAP-1, pCI-eGFP-RAP-2, pCI-eGFP-RAP-2(G12V), pCI-eGFP-RAP-2(S17A)) were generated by replacing Ras in pCI-eGFP-Ras (Yasuda et al., 2006) with hRap2a and *rap-2* cDNAs with *Xho*I and *Bam*HI. pCI-mRFP-RalGDS-mRFP plasmid is a kind gift from Dr. Yasuda. Rap2 and FRET sensor plasmids were mixed in 1:2 ratio and transfected into HeLa cells using Lipofectamine 3000 (ThermoFisher). FLIM was conducted 24 hr after transfection. For expression of FRET sensor in the DA9 neuron, each fusion protein constructs were cloned into *Asc*I and *Kpn*I sites of the pSM vector containing *mig-13* promoter using SLiCE method.

A custom-made two-photon fluorescence lifetime imaging microscope was used as described elsewhere (Murakoshi et al., 2011). Briefly, EGFP-Rap2a was excited with a Ti-sapphire laser (Mai Tai; Spectra-Physics) tuned to 920 nm. The X and Y scanning galvano mirrors (6210 hr; Cambridge Technology) were controlled with ScanImage software (*Pologruto et al., 2003*). EGFP photon signals

were collected an objective lens (60×, 1.0 NA; Olympus) and a photomultiplier tube (H7422-40p; Hamamatsu) placed after a dichroic mirror (FF553-SDi01; Semrock) and emission filter (FF01-510/84; Semrock). A fluorescence lifetime curve was recorded by a time-correlated single-photon-counting board (SPC-150; Becker and Hickl) controlled with the software described previously (Yasuda et al., 2006). For construction of a fluorescence lifetime image, the mean fluorescence lifetime values ($\tau_m$) in each pixel were translated into a color-coded image. We quantified free EGFP-Rap2a and EGFP-Rap2a undergoing FRET (binding fraction) as described elsewhere (Yasuda et al., 2006). Briefly, we calculated the proportion of EGFP undergoing FRET in individual ROIs using the following formula:

$$P_{FRET} = \frac{\tau_{free}(\tau_{free} - \tau_m)}{(\tau_{free} - \tau_{FRET})(\tau_{free} + \tau_{FRET} - \tau_m)} P_{FRET} = \frac{\tau_{free}(\tau_{free} - \tau_m)}{(\tau_{free} - \tau_{FRET})(\tau_{free} + \tau_{FRET} - \tau_m)} \tag{1}$$

where $\tau_{free}$ and $\tau_{FRET}$ are the fluorescence lifetime of free EGFP and EGFP undergoing FRET, respectively.

## Statistics

Prism (GraphPad) software was used for statistical analysis. One-way ANOVA was done and corrected for multiple comparisons with posthoc Tukey's multiple comparisons tests done between all genotypes. Student's t-test was used for pairwise comparison. Sample numbers were pre-determined before conducting statistical analyses.

## Supplemental experimental procedures

The method for qRT-PCR and sequences of primers and repair templates used in this study are available in *Supplementary file 1*.

## Acknowledgements

We are grateful to Donald Moerman and his lab members for generating *rap*-3 mutant strain, providing other mutant strains, sharing reagents and for general discussions. We also thank Ryohei Yasuda for FRET sensor plasmids, Hiroshi Kawabe for human Rap2a cDNA, Richard Ikegami for *plx-2* promoter construct, Harald Hutter and Peri Kurshan for sharing strains, May Dang-Lawson and Michael Gold for unpublished biochemical experiments, Lisa Fernando for the technical support, all Mizumoto lab members and general discussions, Kang Shen, Shaul Yogev and Maulik Patel for comments on the manuscript. Some strains used in this study were obtained from the CGC, which is funded by NIH Office of Research Infrastructure Programs (P40 OD010440) and *C. elegans* gene knock out consortium. Mizumoto Lab is funded by HFSP (CDA-00004/2014), CIHR (PJT-148667), NSERC (RGPIN-2015–04022), CFI-JELF (34722) and Tomizawa Jun-ichi and Keiko Fund for Young Scientist. KM is a recipient of Canada Research Chair and Michael Smith Foundation for Health Research Scholar.

## Additional information

### Funding

| Funder | Grant reference number | Author |
| --- | --- | --- |
| Human Frontier Science Program | CDA-00004/2014 | Kota Mizumoto |
| Canadian Institutes of Health Research | PJT-148667 | Kota Mizumoto |
| Natural Sciences and Engineering Research Council of Canada | RGPIN-2015-04022 | Kota Mizumoto |
| Canada Research Chairs | | Kota Mizumoto |
| Michael Smith Foundation for Health Research | 16869 | Kota Mizumoto |

| Tomizawa Jun-ichi and Keiko Fund of Molecular Biology Society of Japan for Young Scientists | | Kota Mizumoto |
|---|---|---|
| Canada Foundation for Innovation | 34722 | Kota Mizumoto |

The funders had no role in study design, data collection and interpretation, or the decision to submit the work for publication.

### Author contributions
Xi Chen, Formal analysis, Investigation, Visualization; Akihiro CE Shibata, Ardalan Hendi, Mizuki Kurashina, Ethan Fortes, Formal analysis, Validation, Investigation, Visualization, Writing—review and editing; Nicholas L Weilinger, Investigation, Writing—review and editing; Brian A MacVicar, Resources, Supervision; Hideji Murakoshi, Resources, Supervision, Validation, Investigation, Writing—review and editing; Kota Mizumoto, Conceptualization, Data curation, Formal analysis, Supervision, Funding acquisition, Validation, Investigation, Visualization, Writing—original draft, Writing—review and editing

### Author ORCIDs
Brian A MacVicar ![ORCID] http://orcid.org/0000-0003-4596-4623
Kota Mizumoto ![ORCID] http://orcid.org/0000-0001-8091-4483

### Decision letter and Author response
Decision letter https://doi.org/10.7554/eLife.38801.023
Author response https://doi.org/10.7554/eLife.38801.024

## Additional files

### Supplementary files
• Supplementary file 1. Supplementary experimental procedures.
DOI: https://doi.org/10.7554/eLife.38801.019
• Transparent reporting form
DOI: https://doi.org/10.7554/eLife.38801.020

### Data availability
All data generated or analyzed during this study are included in the manuscript and supporting files.

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
