## [Decision Letter]

[Editors’ note: a previous version of this study was rejected after peer review, but the authors submitted for reconsideration. The first decision letter after peer review is shown below.]

Thank you for submitting your work entitled "Rap2 and TNIK control Plexin-dependent synaptic tiling in *C. elegans*" for consideration by *eLife*. Your article has been reviewed by a Senior Editor, a Reviewing Editor, and three reviewers. The reviewers have opted to remain anonymous.

Our decision has been reached after consultation among the reviewers. Based on these discussions and the individual reviews below, we regret to inform you that your work will not be considered further for publication in *eLife*.

All reviewers appreciate the focus on identifying new genes and pathways regulating synaptic tiling. However, to some extent, many of the molecular players implicated in this manuscript have previously been identified in the Sema/Plexin pathway in other contexts (Plexin implicated by this lab in synaptic tiling, Rap is a known target of Plexin signaling, MIG-15/TNIK is a known regulator of Rap). A strength of your manuscript is the description of the roles of all these components in a pathway that regulates synaptic tiling. However, given this, as the reviewers note, there are several gaps/weaknesses in the current analyses tying these components together. The reviewers suggest several experiments to strengthen the work.

Reviewer #1:

This is an interesting study that is the next step in defining the signaling pathway(s) downstream of PLX-1 signaling in regulating tiling of DA8 and DA9 motor neuron synapses. The data supporting a cell-autonomous role for *rap-2* downstream of *plx-1* are convincing, however they are also expected (that at least one *C. elegans* Rap homolog would be downstream) based on other studies of Plexin signaling.

While *mig-15* is clearly doing something interesting in terms of inhibiting synapse formation throughout the worm, the data implicating MIG-15 downstream of PLX-1 RAP-2 signaling is less convincing. The authors' results with respect to MIG-15 feel like the beginning of a new story, and not just a new molecule downstream of Plexins.

Reviewer #2:

In this manuscript, the authors use *C. elegans* DA9 and DA8 neurons as the model to investigate synaptic tilling. The authors describe a potential important role of Rap2 and TNIK in regulating the synaptic tilling of DA neurons. The authors present genetic evidence to support the function of Rap2 and TNIK in this process, however the evidence to link Semaphorin receptors with Rap2 or TNIK was relatively weak.

1) The authors show that both gain-of function and loss-of-function of *rap-2* have similar phenotypes. They use *rap-2* (G12V) as a gain-of-function mutation for RAP-2, is this mutation really cause gain-of-function in *C. elegans* neurons? The authors need either cite published results or test it by themselves.

2) The authors use the co-localzation of mCherry::RAB-3 with CLA-1 as evidence to show those synapses are "functional synapses". To really reach the conclusion that those synapses are functional synapses, one needs to show either those synapses have similar structures as "normal" synapses, or to show those synapses have similar ability to release neuronal transmitters. I would suggest the authors either use other term (rather than functional synapse) to describe the phenotype or carry out EM or electrophysiology studies to prove that those synapses are functional synapses. It will also be helpful if the authors can test other synaptic markers such as SNB-1, SID-1, SYD-2, to confirm the results.

3) The link between PLX-1 with Rap2 activation was wake. The authors show the activation of RAP-2 has correlation with the localization of PLX-1, but this doesn't mean the activation of RAP-2 is directly regulated by the PLX-1 or depend the activation of PLX-1. Since this is one of the major conclusions of the paper, some direct evidence is needed to prove the activation of RAP2 is indeed locally and directly regulated by PLX-1 and the PLX-1 ligand. Otherwise, the authors need to revise their conclusion.

4) As shown in Figure 7, *mig-15*(OE) seems to affect the tilling border DA8/DA9/. Does *mig-15*(OE) also affect the localization of PLX-1 and the local activation of Rap2?

Reviewer #3:

Chen et al., identify the small GTPase *rap-2* and its effector *mig-15*/TNIK as novel regulators of synaptic tiling in DA8/DA9 motor neurons in *C. elegans*, and further show that MIG-15 inhibits synapse formation in multiple neuron types. Mizumoto et al. (2013) previously showed that Semaphorin/Plexin signaling regulates branched actin via the Plexin GAP domain to control synaptic tiling between DA8 and DA9. Semaphorin/Plexin signaling is known to inhibit synapse formation (Matthes, 1995, Tran, 2009, Duan, 2014, O'Connor, 2009), however the intracellular signaling mechanisms that connect Plexin to synapse development are not known. Based on published data, cited by the authors, Plexins are known to inhibit the Rap family of small GTPases to regulate actin remodeling (Wang, 2012, Wang, 2013), and TNIK family kinases are known downstream effectors of Rap-GTP (Hussain, 2010, Kawabe, 2010, Nonaka, 2008). Thus, the authors test if *C. elegans* Rap family members and *mig-15*/TNIK regulate synaptic tiling downstream of PLX-1. Chen et al. found that *rap-2* and *mig-15* mutants have strong defects in synaptic tiling. The authors use FLIM-based FRET to show that RAP2A is inhibited locally at the synaptic tiling border between DA8 and DA9, and this local effect is abrogated in *plx-1* mutants. They show that *rap-2* functions in DA9 neurons to regulate synaptic tiling and use genetic double mutants to show that *plx-1* and *rap-2* act in the same pathway. Chen et al., identify the Rap-GTP effector kinase, *mig-15*/TNIK as a novel regulator of synaptic tiling and, interestingly, as a general inhibitor of synapse formation throughout the nervous system.

1) The images showing PLX-1::GFP localization to the tiling border are not clear. Mizumoto et al., previously showed that PLX-1::GFP is highly expressed and diffuse in the ventral dendrite and asynaptic regions of the axon including in the axon commissure and also anterior to the synaptic tiling border, with some dimmer puncta within the synaptic region (Neuron 2013). In this manuscript (Figure 3A-B), PLX-1::GFP is much dimmer and punctate at the tiling border. Better images and quantification of PLX-1::GFP distribution, as previously performed, would improve this data.

2) In Figure 3F-G, the authors show that local inhibition of Rap2A at the tiling border does not happen in the absence of *plx-1* and conclude that the data "strongly suggest that Plexin localized at the anterior edge of the DA9 synaptic domain locally inactivates Rap2A GTPase to delineate the synaptic tiling border in DA9." Since *plx-1* mutants are missing PLX-1 in the whole animal including throughout DA9, a better experiment might be to rescue the mutant with PLX-1(ΔSema)::GFP, which the authors previously showed was mislocalized throughout DA9, and show that mislocalized, inactive PLX-1 does not rescue the local inactivation of Rap2A. And importantly, does GAP-deficient PLX-1(RA)::GFP, which the authors previously showed was properly localized to the tiling border, show no change in local Rap2A activity?

3) Can the authors show that there are no subtle defects in DA8/DA9 axon contact in *mig-15* mutants that could explain the stronger synaptic tiling defects? The authors show that *mig-15* mutants have strong defects in axon outgrowth and branching in about 50% of the animals (Figure 5—figure supplement 1). In addition, *mig-15* mutants have a more dramatic DA8/DA9 overlap phenotype (50-55um overlap in Figure 4) compared to *plx-1* or *rap-2* mutants (which have ~30um overlap in Figure 1). The authors previously showed (Mizumoto et al., 2013) that in axon guidance mutants unc-34 and unc-129, DA9 is misguided and does not make axon contacts in the dorsal cord leading to strong DA8/DA9 synaptic tiling overlap defects (up to 50um). Thus, it is not clear if the DA8/DA9 overlap defects observed in *mig-15* mutants are primarily due to axon guidance defects (i.e. more subtle defects where DA8 and DA9 do not contact each other) or to a more direct role for *mig-15* in synaptic tiling.

4) It is not clear whether *mig-15* acts upstream, downstream or in parallel to plx-1 and *rap-2* to regulate synaptic patterning. If MIG-15 acts downstream of PLX-1, and Rap-2-GTP binds and activates TNIK/MIG-15 to inhibit synapse formation, then MIG-15(OE) should inhibit synapse formation independent of *plx-1* and *rap-2*. Instead, Figure 7E appears to show that the effects of MIG-15(OE) on DA9 synapses are suppressed by *plx-1* or *rap-2*. The interpretation of these results should be clarified. Also, it would be helpful if *plx-1* and *rap-2* single mutants are shown for comparison in Figure 7E to determine whether MIG-15 functions upstream of *plx-1* and *rap-2* or in parallel.

5) Have the authors tested if there are any functional/ behavioral consequences to defects in synaptic tiling between DA8 and DA9?

[Editors’ note: what now follows is the decision letter after the authors submitted for further consideration.]

Thank you for submitting your article "Rap2 and TNIK control Plexin-dependent synaptic tiling in *C. elegans*" for consideration by *eLife*. Your article has been reviewed by K VijayRaghavan as the Senior Editor, a Reviewing Editor, and three reviewers. The reviewers have opted to remain anonymous.

The reviewers have discussed the reviews with one another and the Reviewing Editor has drafted this decision to help you prepare a revised submission.

While all reviewers appreciate the additional data and analyses presented in this manuscript, and the extent to which previous comments have been addressed, the major remaining concern is about the placement of MIG-15 in the PLX-1 pathway. The data supporting the notion that MIG-15 is downstream of PLX-1 are weak. While this issue is addressed in part in Figure 8, the manuscript should be revised to address this concern. Specifically, the manuscript should overtly state the limitations of the genetic and phenotypic analyses and only speculate about the mechanistic relationship between MIG-15 and PLX-1.

---

## [Author Response]

[Editors’ note: the author responses to the first round of peer review follow.]

Reviewer #1:This is an interesting study that is the next step in defining the signaling pathway(s) downstream of PLX-1 signaling in regulating tiling of DA8 and DA9 motor neuron synapses. The data supporting a cell-autonomous role for rap-2 downstream of plx-1 are convincing, however they are also expected (that at least one C. elegans Rap homolog would be downstream) based on other studies of Plexin signaling.

We appreciate the reviewer’s enthusiasm in our work. While it is indeed already shown that Plexin functions as a RapGAP, our work is the first to show the role of Plexin/Rap signaling in synapse pattern formation in vivo. More importantly, using FLIM, we show for the first time that Plexin controls the spatial activity of Rap2 small GTPase within the specific axonal segment. With our additional data to support this observation as described below, we believe that this work provides a significant advance in our knowledge of the Plexin/Rap2 signaling pathway in synapse pattern formation.

While mig-15 is clearly doing something interesting in terms of inhibiting synapse formation throughout the worm, the data implicating MIG-15 downstream of PLX-1 RAP-2 signaling is less convincing. The authors' results with respect to MIG-15 feel like the beginning of a new story, and not just a new molecule downstream of Plexins.

We absolutely agree with the reviewer’s comment. This work is a beginning of our next exciting challenge in understanding the roles of TNIK/MIG-15 in inhibiting presynaptic development. While the larger synaptic tiling defects we observed in *mig-15* mutants clearly suggests that *mig-15* functions in multiple pathways as we discussed in the Discussion section, we still believe that *mig-15* functions downstream of *plx-1* in synaptic tiling. The critical observation we added this time was PLX-1::GFP localization was not affected in the *mig-15(rh148)* mutant background, while ectopic synapses are formed beyond the PLX-1::GFP patch at the putative synaptic tiling border. This data strongly suggests that *mig-15* controls synapse patterning downstream of PLX-1. We therefore believe that it is appropriate to include *mig-15* as a Plexin/Rap2 signaling component in this manuscript.

Reviewer #2:In this manuscript, the authors use C. elegans DA9 and DA8 neurons as the model to investigate synaptic tilling. The authors describe a potential important role of Rap2 and TNIK in regulating the synaptic tilling of DA neurons. The authors present genetic evidence to support the function of Rap2 and TNIK in this process, however the evidence to link Semaphorin receptors with Rap2 or TNIK was relatively weak.1) The authors show that both gain-of function and loss-of-function of rap-2 have similar phenotypes. They use rap-2(G12V) as a gain-of-function mutation for RAP-2, is this mutation really cause gain-of-function in C. elegans neurons? The authors need either cite published results or test it by themselves.

As we showed in Figure 1, expression of *rap-2(G12V)* in wild-type background caused synaptic tiling defects, suggesting that *rap-2(G12V)* is a gain-of-function mutant. We observed no synaptic tiling defects when we injected wild-type *rap-2* under the same *unc-4* promoter in wild-type. This confirms that the synaptic tiling defect observed in animals expressing *rap-2(G12V)* is not due to the over-expression but rather the G12V mutation itself. This data is included in the revised Figure 1. We also rephrased ‘over-expression’ to ‘expression’ in the following sentence:

“Expression of a constitutively GTP-bound form of *rap-2* (G12V) under the A-type neuron specific promoter, *Punc-4*, elicited a similar synaptic tiling defect as *plx-1* mutants (Figures 1C and 1G). Expression of wild type *rap-2* under the unc-4 promoter did not affect the synaptic tiling pattern, suggesting that G12V mutation but not over-expression of *rap-2* caused the synaptic tiling defect (Figure 1G).*”*

hRap2A 1 MREYKVVVLGS**G**GVGKSALTVQFVTGTFIEKYDPTIEDFYRKEIEVDSSPSVLEILDTAG 60

MRE+KVVVLGSGGVGKSALTVQFV+ TFIEKYDPTIEDFYRKEIEVD PSVLEILDTAG

RAP-2 1 MREFKVVVLGS**G**GVGKSALTVQFVSSTFIEKYDPTIEDFYRKEIEVDGQPSVLEILDTAG 60

In addition, considering the high sequence identity between *C. elegans* RAP-2 and human Rap2A (please see above sequence comparison), and the fact that G12V mutation is widely used as a constitutively active mutant for many small GTPases including Ras, Rac, Cdc42, Rap (including RAP-1 in *C. elegans:* MBC 2005 16(1): 106–116), we believe that *C. elegans* RAP-2 (G12V) is also a gain-of-function mutant.

We added the following sentence to explain G12V mutation generally used in Rap GTPase field.

The G12V mutant is widely used as a constitutively GTP-form of small GTPases including mammalian Rap2A and *C. elegans* RAP-1 (Kawabe et al., 2010; Pellis van Berkal et al., 2005).

We also conducted GST-RalGDS(RBD) pull-down assay in 293T cell line, which is a well-established assay to detect GTP-form of Rap small GTPase (McLeod et al., JBC 1998; Lin et al., Oncogene 2010) and found that *C. elegans* RAP-2(G12V) was coimmunoprecipitated with RalGDS (See Author response image 1). We were a little bit hesitant to put this figure in the manuscript since we did not detect *C. elegans* RAP-2 in the lysate due to low expression in 3 replicates (right gel), which hindered us from conducting quantitative analysis. Nevertheless, this data is also consistent with the idea that *C. elegans* RAP-2(G12V) is a constitutively active form.

The FLIM-FRET experiment we showed in response to reviewer1’s comment 2 also suggests that RAP-2(G12V) is a constitutively GTP-bound form.

2) The authors use the co-localzation of mCherry::RAB-3 with CLA-1 as evidence to show those synapses are "functional synapses". To really reach the conclusion that those synapses are functional synapses, one needs to show either those synapses have similar structures as "normal" synapses, or to show those synapses have similar ability to release neuronal transmitters. I would suggest the authors either use other term (rather than functional synapse) to describe the phenotype or carry out EM or electrophysiology studies to prove that those synapses are functional synapses. It will also be helpful if the authors can test other synaptic markers such as SNB-1, SID-1, SYD-2, to confirm the results.

We now examined another active zone markers, UNC-10::TdTomato, which colocalized with RAB-3 synaptic vesicle markers in the *plx-1, rap-2* and *mig-15* mutants. The representative images are in Figure S3. We also re-phrased ‘functional synapses’ to ‘bona fide synapses’.

3) The link between PLX-1 with Rap2 activation was wake. The authors show the activation of RAP-2 has correlation with the localization of PLX-1, but this doesn't mean the activation of RAP-2 is directly regulated by the PLX-1 or depend the activation of PLX-1. Since this is one of the major conclusions of the paper, some direct evidence is needed to prove the activation of RAP2 is indeed locally and directly regulated by PLX-1 and the PLX-1 ligand. Otherwise, the authors need to revise their conclusion.

We appreciate the reviewer’s comment. This is also raised by the reviewer 3. As reviewer 3 suggested, we conducted additional FLIM experiments using two mutant PLX-1 rescuing constructs; one cannot be activated by Semaphorin; the other has no GAP activity but can localize normally (please see our response to reviewer 3, comment 2 for details). Neither of them rescued the local Rap2 inactivation pattern. These data strongly support our model that PLX-1 controls local RAP-2 activity.

Regarding whether PLX-1 directly regulates Rap2, we do not have clear evidence for this and therefore did not mention that it was direct. However, we think it is highly likely since structural and biochemical experiments using mammalian Plexin nicely showed that Plexin is a RapGAP. To clarify our conclusion, we added the following sentence.

“While we do not fully exclude the possibility that PLX-1 indirectly regulated local Rap2 activity, together with the biochemical evidence that mammalian Plexin acts as RapGAP (Wang et al., 2012; 2013), these data strongly suggests that Plexin localized at the anterior edge of the DA9 synaptic domain locally inactivates Rap2 GTPase to delineate the synaptic tiling border in DA9.”

4) As shown in Figure 7, mig-15(OE) seems to affect the tilling border DA8/DA9/. Does mig-15(OE) also affect the localization of PLX-1 and the local activation of Rap2?

We indeed observed the posterior shift of PLX-1::GFP in animals expressing *mig-15.* A representative image is now included in Figure 3.Due to the variability of the phenotype caused by *mig-15* overexpression from an extrachromosomal array, we were not able to quantify the shift of PLX-1::GFP. For the same variability issue, we were not able to conduct FLIM experiment.

Reviewer #3:[…] 1) The images showing PLX-1::GFP localization to the tiling border are not clear. Mizumoto et al., previously showed that PLX-1::GFP is highly expressed and diffuse in the ventral dendrite and asynaptic regions of the axon including in the axon commissure and also anterior to the synaptic tiling border, with some dimmer puncta within the synaptic region (Neuron 2013). In this manuscript (Figure 3A-B), PLX-1::GFP is much dimmer and punctate at the tiling border. Better images and quantification of PLX-1::GFP distribution, as previously performed, would improve this data.

We appreciate the reviewer’s suggestion. We re-set the imaging condition for PLX-1::GFP so that the PLX-1::GFP patch is now more obvious than the original ones. We also added the quantification of the signals in Figure 3.

2) In Figure 3F-G, the authors show that local inhibition of Rap2A at the tiling border does not happen in the absence of plx-1 and conclude that the data "strongly suggest that Plexin localized at the anterior edge of the DA9 synaptic domain locally inactivates Rap2A GTPase to delineate the synaptic tiling border in DA9." Since plx-1 mutants are missing PLX-1 in the whole animal including throughout DA9, a better experiment might be to rescue the mutant with PLX-1(ΔSema)::GFP, which the authors previously showed was mislocalized throughout DA9, and show that mislocalized, inactive PLX-1 does not rescue the local inactivation of Rap2A. And importantly, does GAP-deficient PLX-1(RA)::GFP, which the authors previously showed was properly localized to the tiling border, show no change in local Rap2A activity?

We again appreciate the reviewer’s insightful suggestion. We conducted the suggested experiments and indeed observed no rescue in Rap2 activity pattern when PLX-1(ΔSema) or PLX-1(RA) was expressed in DA9. On the other hand, expression of wildtype PLX-1 nicely recreated the region with low Rap2 activity in the *plx-1* mutant background. This observation strengthens our conclusion that PLX-1 locally inhibits Rap2 activity at the synaptic tiling border.

We believe this experiment also answered the question 3 from reviewer 2. The new data is now added to Figure 4.

3) Can the authors show that there are no subtle defects in DA8/DA9 axon contact in mig-15 mutants that could explain the stronger synaptic tiling defects? The authors show that mig-15 mutants have strong defects in axon outgrowth and branching in about 50% of the animals (Supplemental Figure 4). In addition, mig-15 mutants have a more dramatic DA8/DA9 overlap phenotype (50-55um overlap in Figure 4) compared to plx-1 or rap-2 mutants (which have ~30um overlap in Figure 1). The authors previously showed (Mizumoto et al., 2013) that in axon guidance mutants unc-34 and unc-129, DA9 is misguided and does not make axon contacts in the dorsal cord leading to strong DA8/DA9 synaptic tiling overlap defects (up to 50um). Thus, it is not clear if the DA8/DA9 overlap defects observed in mig-15 mutants are primarily due to axon guidance defects (i.e. more subtle defects where DA8 and DA9 do not contact each other) or to a more direct role for mig-15 in synaptic tiling.

In our previous paper, we showed that the PLX-1::GFP localization is dependent on the axon-axon interaction. In the axon guidance mutants in which axon-axon interaction between DA8 and DA9 is disrupted, PLX-1::GFP was distributed evenly throughout the axon. In *mig-15* mutants, PLX-1::GFP localization was largely unaffected, suggesting that the synaptic tiling defect in *mig-15* is not due to the loss of axon-axon interaction between DA8 and DA9. Now this data is included in Figure 3. We added the following sentence in the revised manuscript.

“The PLX-1::GFP patch at the putative synaptic tiling border was unaffected in *mig-15* mutants, even though the position of the PLX-1::GFP patch has shifted slightly posteriorly compared with wild-type animals (Figure 3C), suggesting that *mig-15* acts downstream of PLX-1 in regulating synaptic tiling.”

In addition, we did not observe severe synaptic tiling defect in *cdh-4(rh310)*, which showed axon defasciculation phenotype in the dorsal nerve cord neurons (Schmitz et al., 2008). The overlap between DA8 and DA9 synaptic domains was 4.6 µm (n=21, SEM ± 1.02). This observation also supports the idea that the large overlap between DA8 and DA9 synaptic domains in *mig-15* mutants is not due to the loss of axon-axon interaction. We included the following sentence:

“In addition, we did not observe significant synaptic tiling defects in *cdh-4(rh310)* mutants (4.6 ± 1.02 µm, n=21), which exhibits axon defasciculation phenotype in the dorsal nerve cord neurons (Schmitz et al., 2008). These data suggest that the synaptic tiling defect in the *mig-15* mutants is not a secondary effect of axon outgrowth and guidance.”

4) It is not clear whether mig-15 acts upstream, downstream or in parallel to plx-1 and rap-2 to regulate synaptic patterning. If MIG-15 acts downstream of PLX-1, and Rap-2-GTP binds and activates TNIK/MIG-15 to inhibit synapse formation, then MIG-15(OE) should inhibit synapse formation independent of plx-1 and rap-2. Instead, Figure 7E appears to show that the effects of MIG-15(OE) on DA9 synapses are suppressed by plx-1 or rap-2. The interpretation of these results should be clarified. Also, it would be helpful if plx-1 and rap-2 single mutants are shown for comparison in Figure 7E to determine whether MIG-15 functions upstream of plx-1 and rap-2 or in parallel.

We believe that *mig-15* has two roles. First, *mig-15* acts downstream of *plx-1* and *rap-2* in synapse patterning (or synaptic tiling). We propose this based on the synaptic tiling defect (but normal PLX-1 localization) in *mig-15* mutants. Second, *mig-15* functions as a key negative regulator of synapse number. Since we did not observe a significant increase in synapse numbers in *plx-1* or *rap-2* mutants, the role of *mig-15* in controlling synapse number is independent of Plexin signaling pathway. Indeed, the reduction of synapse number in *mig-15-*overexpressing animals is not suppressed in *plx-1* or *rap-2* mutants. We added the following sentence in the revised manuscript to clarify the relationship between Plex/Rap pathway and *mig-15.*

“Synapse number was not different in mig-15(OE) and *rap-2*(gk11); mig-15(OE) animals, suggesting that the role of mig-15 in inhibiting synapse number is not dependent on Plexin/Rap2 signaling pathway (Figure S6).”

5) Have the authors tested if there are any functional/ behavioral consequences to defects in synaptic tiling between DA8 and DA9?

All synaptic tiling mutants we have isolated so far, except for *mig-15* mutantswhich are sick and hence do not move well, exhibit superficially normal locomotion. This is probably due to the functional redundancy in the motor circuit, in which subtle synaptic pattern defect could be compensated by other mechanisms. Most of the *C. elegans* mutants with synapse patterning defects indeed show minimal locomotion defects, and hence have been missed from the traditional locomotion-based genetic screenings.

[Editors' note: the author responses to the re-review follow.]

The reviewers have discussed the reviews with one another and the Reviewing Editor has drafted this decision to help you prepare a revised submission.While all reviewers appreciate the additional data and analyses presented in this manuscript, and the extent to which previous comments have been addressed, the major remaining concern is about the placement of MIG-15 in the PLX-1 pathway. The data supporting the notion that MIG-15 is downstream of PLX-1 are weak. While this issue is addressed in part in Figure 8, the manuscript should be revised to address this concern. Specifically, the manuscript should overtly state the limitations of the genetic and phenotypic analyses and only speculate about the mechanistic relationship between MIG-15 and PLX-1.

We agree with the reviewers’ comments.

In the revised manuscript, we emphasized that our works revealed the genetic relationship between *mig-15* and *plx-1* by adding ‘genetically’ in the following sentences.

Abstract:

“Here, we show that Rap2 GTPase (*rap-2*) and its effector, TNIK (*mig-15*), act genetically downstream of Plexin (*plx-1*) to restrict presynaptic assembly and form tiled synaptic innervation in *C. elegans*.”

Introduction:

“Here, we report that *rap-2*, a *C. elegans* ortholog of human Rap2A, and its effector kinase *mig-15* (TNIK: *Traf2*- and Nck-interacting kinase) act genetically downstream of PLX-1 to regulate synaptic tiling.”

In the following sentence, *rap-2* was replaced to *plx-1* as we did not show that *mig-15* acts downstream of *rap-2* in synaptic tiling.

Discussion section:

“We showed that proper synapse patterning requires both GDP- and GTP-forms of RAP-2. Considering that PLX-1 regulates the spatial distribution of RAP-2 activity and *mig-15* acts genetically downstream of *plx-1* in synaptic tiling, RAP-2 may also locally regulate MIG-15 (TNIK).”

The following sentences were added to describe the limitation of our analyses and the needs of future biochemical analysis to fully understand the mechanisms of PLX-1/Rap2/TNIK signaling in synapse pattern formation.

Discussion section:

“Due to the pleiotropic phenotype of the *mig-15* mutants, our genetic and phenotypic analyses of *mig-15* did not exquisitely reveal the mechanistic relationship between PLX-1 and MIG-15 in synaptic tiling regulation. Further biochemical studies of MIG-15 regulation by Plexin/Rap2 in synaptic tiling will elucidate the molecular mechanisms that underlie the role of MIG-15/TNIK in synapse pattern formation.”